# BK channel properties correlate with neurobehavioral severity in three *KCNMA1*-linked channelopathy mouse models

**Su Mi Park[1], Cooper E Roache[1], Philip H Iffland II[2], Hans J Moldenhauer[1], Katia K Matychak[1], Amber E Plante[1], Abby G Lieberman[3], Peter B Crino[2], Andrea Meredith[1]***

[1]Department of Physiology, University of Maryland School of Medicine, Baltimore, United States; [2]Department of Neurology, University of Maryland School of Medicine, Baltimore, United States; [3]Department of Pharmacology, University of Maryland School of Medicine, Baltimore, United States

*For correspondence:
ameredith@som.umaryland.edu

**Competing interest:** The authors declare that no competing interests exist.

**Abstract** KCNMA1 forms the pore of BK $K^+$ channels, which regulate neuronal and muscle excitability. Recently, genetic screening identified heterozygous *KCNMA1* variants in a subset of patients with debilitating paroxysmal non-kinesigenic dyskinesia, presenting with or without epilepsy (PNKD3). However, the relevance of *KCNMA1* mutations and the basis for clinical heterogeneity in PNKD3 has not been established. Here, we evaluate the relative severity of three *KCNMA1* patient variants in BK channels, neurons, and mice. In heterologous cells, $BK^{N999S}$ and $BK^{D434G}$ channels displayed gain-of-function (GOF) properties, whereas $BK^{H444Q}$ channels showed loss-of-function (LOF) properties. The relative degree of channel activity was $BK^{N999S} > BK^{D434G} > WT > BK^{H444Q}$. BK currents and action potential firing were increased, and seizure thresholds decreased, in $Kcnma1^{N999S/WT}$ and $Kcnma1^{D434G/WT}$ transgenic mice but not $Kcnma1^{H444Q/WT}$ mice. In a novel behavioral test for paroxysmal dyskinesia, the more severely affected $Kcnma1^{N999S/WT}$ mice became immobile after stress. This was abrogated by acute dextroamphetamine treatment, consistent with PNKD3-affected individuals. Homozygous $Kcnma1^{D434G/D434G}$ mice showed similar immobility, but in contrast, homozygous $Kcnma1^{H444Q/H444Q}$ mice displayed hyperkinetic behavior. These data establish the relative pathogenic potential of patient alleles as N999S>D434G>H444Q and validate $Kcnma1^{N999S/WT}$ mice as a model for PNKD3 with increased seizure propensity.

## Editor's evaluation

This study is of broad interest to neuroscientists interested in membrane excitability and translational biologists and physicians eager for robust animal models for disorders involving mutations in the KCNMA gene, such as paroxysmal nonkinesigenic dyskinesia PNKD3. Here, phenotypes of mouse models of three of the more common patient disease-related mutations in KCNMA are evaluated for similarities to patient phenotypes. This work establishes that BK channel mutations linked to human neurological disease can, on their own, cause similar pathology in mice, and it also begins to provide neurological bases for the associated behavioral deficits. Importantly, one of the mutant alleles expressed in mice most closely phenocopies the patient phenotype, which is rescued with a new treatment for PNKD3 in KCNMA1-N999S patients, further validating it as an important animal model for studies seeking therapeutic treatments for the resulting debilitating disease moving forward.

**eLife digest** So far, only 70 patients around the world have been diagnosed with a newly identified rare syndrome known as *KCNMA1*-linked channelopathy. The condition is characterised by seizures and abnormal movements which include frequent 'drop attacks', a sudden and debilitating loss of muscle control that causes patients to fall without warning.

The disease is associated with mutations in the gene for KCNMA1, a member of a class of proteins important for controlling nerve cell activity and brain function. However, due to the limited number of people affected by the condition, it is difficult to link a particular mutation to the observed symptoms; the basis for the drop attacks therefore remains unknown. Park et al. set out to 'model' *KCNMA1*-linked channelopathy in the laboratory, in order to determine which mutations in the *KCNMA1* gene caused these symptoms.

Three groups of mice were each genetically engineered to carry either one of the two most common mutations in the gene for KCNMA1, or a very rare mutation associated with the movement symptoms. Behavioural experiments and studies of nerve cell activity revealed that the mice carrying mutations that made the KCNMA1 protein more active developed seizures more easily and became immobilized, showing the mouse version of drop attacks. Giving these mice the drug dextroamphetamine, which works in some human patients, stopped the immobilizing attacks altogether.

These results show for the first time which specific genetic changes cause the main symptoms of *KCNMA1*-linked channelopathy. Park et al. hope that this knowledge will deepen our understanding of this disease and help develop better treatments.

## Introduction

*KCNMA1*-linked channelopathy encompasses an array of neurological symptoms associated with clinical detection of a *KCNMA1* variant. Affected individuals typically present with epilepsy and/or dyskinesia, but also have other disorders including ataxia, developmental delay, intellectual disability, and brain and structural abnormalities (*Bailey et al., 2019*; *Liang et al., 2019*; *Miller et al., 2021*). The basis for these symptoms is not mechanistically established but is likely similar to other neurological channelopathies involving direct or indirect changes in neuronal excitability leading to excitation-inhibition imbalance (*Benatar, 2000*; *Menezes et al., 2020*). *KCNMA1* genotype-phenotype correlation is an active area of investigation with >40 variants identified in this patient population to date (*Miller et al., 2021*, and ALM unpublished data). Since most variants arise de novo in a single heterozygous proband, whether '*KCNMA1* channelopathy' is a bona fide monogenic disorder, or results from intergenic and developmental interactions, is not well understood. Animal models for the most common variants are needed to validate genotype-phenotype associations and to investigate disease mechanisms and manifestations over lifespan (*MacArthur et al., 2014*).

*KCNMA1* encodes the 'Big K$^+$' (BK) channel, activated by voltage and intracellular Ca$^{2+}$ (*Figure 1*). BK currents are prominent in the central nervous system and smooth muscle (*Bailey et al., 2019*; *Contet et al., 2016*; *Latorre et al., 2017*). Neuronal BK channels regulate action potential repolarization and fast afterhyperpolarizations (fAHP) to set firing rates (*Gu et al., 2007*; *Montgomery and Meredith, 2012*; *Sah and Faber, 2002*; *Shao et al., 1999*) and neurotransmission (*Golding et al., 1999*; *Raffaelli et al., 2004*; *Sailer et al., 2006*; *Tazerart et al., 2022*). KCNMA1 knockout mice (*Kcnma1*$^{-/-}$) show prominent smooth muscle, neurobehavioral, and locomotor deficits, associated with widespread alterations in cellular excitability (MGI:99923; *Bailey et al., 2019*; *Meredith et al., 2004*; *Sausbier et al., 2005*; *Sausbier et al., 2004*). However, *Kcnma1*$^{-/-}$ mice do not overtly exhibit *KCNMA1*-linked channelopathy symptoms. Moreover, the largest cohort of clinically distinguishable patients harbor gain-of-function (GOF), rather than loss-of-function (LOF), alleles with respect to BK channel activity (*Miller et al., 2021*).

Two GOF *KCNMA1* variants, D434G and N999S, account for half of the patient population (*Bailey et al., 2019*; *Miller et al., 2021*). Both variants cause BK channel activation at more negative membrane potentials, speed activation, and slow deactivation (*Diez-Sampedro et al., 2006*; *Du et al., 2005*; *Li et al., 2018*; *Moldenhauer et al., 2020a*; *Wang et al., 2009*; *Yang et al., 2010*). The majority of individuals harboring D434G and N999S variants present with paroxysmal non-kinesigenic dyskinesia (PNKD type 3; OMIM #609446), characterized by varying degrees of negative motor phenomena

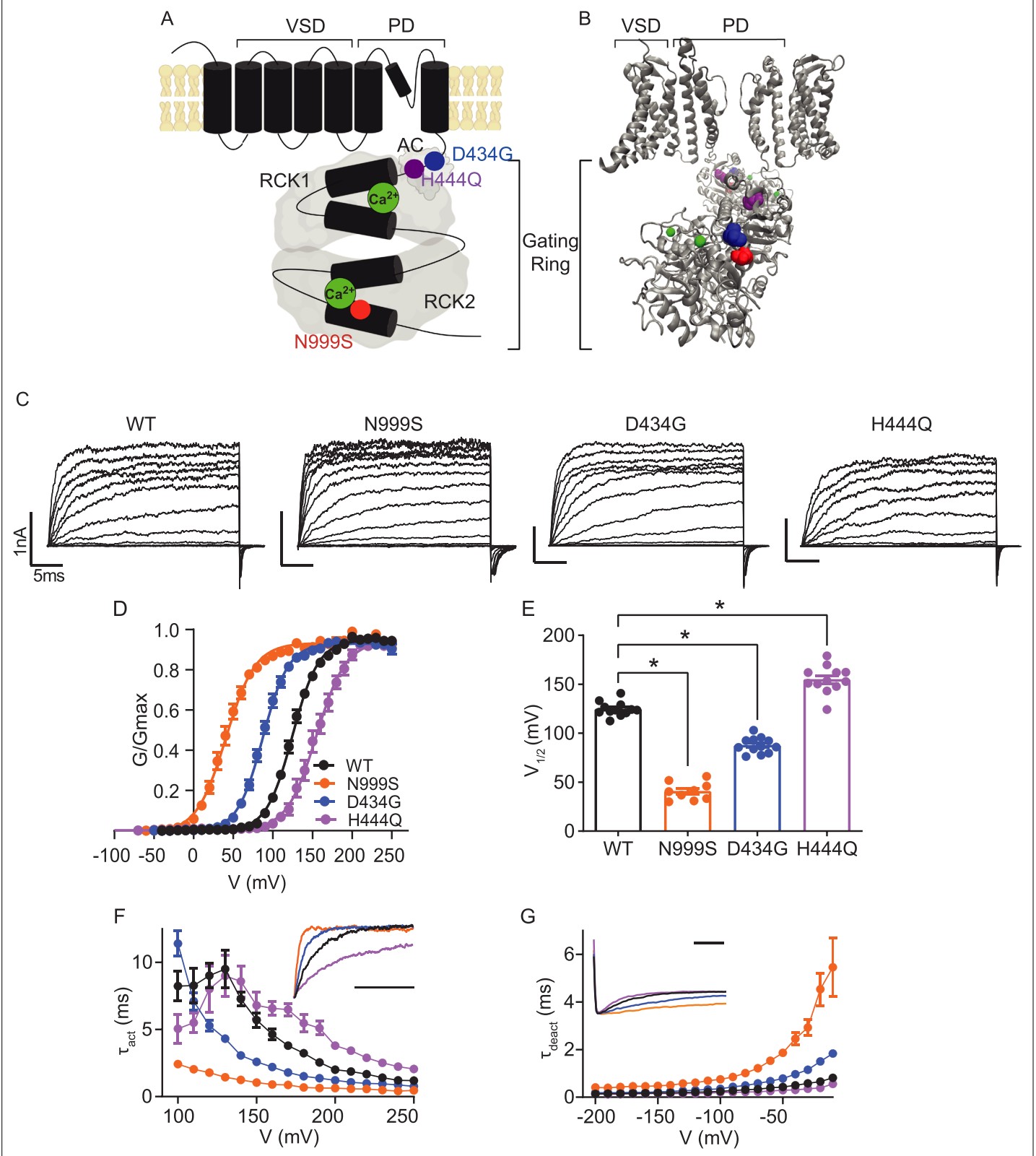

**Figure 1.** Location and consequence of *KCNMA1* variants in the BK K+ channel. (**A**) *KCNMA1* forms the homotetrameric BK channel. Each α subunit is comprised of seven transmembrane domains (S0–S6) and an intracellular gating ring with hydrophobic segments (S7–S10, black). Pore (+) opening and closing is regulated by voltage-sensitive residues in S2–S4 (VSD), the AC domain (βA to αC), and two Regulators of Conductance of Potassium (RCK) domains in the gating ring (gray), each containing a Ca2+ binding site (green) (*Yang et al., 2015*; *Giraldez and Rothberg, 2017*). (**B**) BK channel

*Figure 1 continued on next page*

*Figure 1 continued*

structure showing two opposing subunits with $Ca^{2+}$ bound in the gating ring (PDB 6V38). H444Q (purple) and D434G (blue) are located within the βB-αB and αA and βB of the AC domain, respectively, a region within RCK1 affecting $Ca^{2+}$-dependent gating (*Du et al., 2005*; *Tao and MacKinnon, 2019*). N999S (red) is located at the helix bend in the middle of the S10 domain within RCK2 (*Tao and MacKinnon, 2019*). (**C**) Representative inside-out patch-clamp recordings from $BK^{WT}$, $BK^{N999S}$, $BK^{D434G}$, $BK^{H444Q}$ channels expressed in HEK293 cells. Macroscopic BK currents were recorded in symmetrical $K^+$ and 1 μM intracellular $Ca^{2+}$ by holding patches at −100 mV, stepping from −100 to 250 mV for 30 ms, followed by a tail step −100 mV for 15 ms. Scale bars: 1 nA, 5 ms. (**D**) Normalized conductance-voltage (G-V) relationships fit with Boltzmann functions (solid lines). There was no change in the slope factor (**z**) for any of the variants (p=0.06, one-way ANOVA). $BK^{WT}$ (n=12), $BK^{N999S}$ (n=9), $BK^{D434G}$ (n=12), and $BK^{H444Q}$ (n=12). (**E**) Voltage of half-maximal activation ($V_{1/2}$) obtained from Boltzmann fits for individual patches. *p<0.0001. One-way ANOVA with Dunnett's post hoc. (**F**) Activation time constants ($\tau_{act}$). $BK^{N999S}$ and $BK^{D434G}$ channels had decreased $\tau_{act}$ compared to $BK^{WT}$, either across all voltage steps (mixed effects model for repeated measures with Bonferroni post hoc, p<0.01) or above 120 mV (p<0.05), respectively. At lower voltages, $BK^{D434G}$ channels were more steeply voltage dependent, but did not exceed the fast activation time constants of $BK^{N999S}$ channels. $BK^{H444Q}$ channels had increased $\tau_{act}$ compared to $BK^{WT}$ between 160 and 250 mV (p<0.05). *Inset*: Representative current traces from 170 mV step, scaled to the maximal current to illustrate activation timecourse (x-axis scale bar: 10 ms). $\tau_{act}$ for $BK^{WT}$ currents was 3.8±0.3 ms, while $BK^{N999S}$ and $BK^{D434}$ currents activated faster (0.9±9.1 and 1.8±0.1 ms, respectively) and $BK^{H444Q}$ activated slower (6.5±0.5 ms). (**G**) Deactivation time constants ($\tau_{deact}$). $BK^{N999S}$ and $BK^{D434G}$ channels had increased $\tau_{deact}$ compared to $BK^{WT}$, across all voltage steps (mixed effects model for repeated measures with Bonferroni post hoc, p<0.01), with the exception of −160 (p>0.05), respectively. $BK^{H444Q}$ channels had decreased $\tau_{deact}$ compared to $BK^{WT}$ between −190 mV and between −140 and −20 mV (p<0.05). *Inset*: Representative current traces from −20 mV step, scaled to the maximal current to illustrate deactivation timecourse (x-axis scale bar: 10 ms). $\tau_{deact}$ for $BK^{WT}$ currents was 0.7±0.01 ms, while $BK^{N999S}$ and $BK^{D434}$ currents deactivated slower (4.5±0.7 and 1.5±0.1 ms, respectively) and $BK^{H444Q}$ deactivated more quickly (0.4±0.01 ms). Data are presented as mean ± SEM. Additional data on the effects of stimulants on $BK^{WT}$ and $BK^{N999S}$ channels appears in *Figure 1—figure supplement 1*.

The online version of this article includes the following source data and figure supplement(s) for figure 1:

**Source data 1.** Data file for *Figure 1D–G*.

**Figure supplement 1.** Sequences for CRISPR/Cas9 editing of the mouse *Kcnma1* gene.

**Figure supplement 2.** Effects of lisdexamfetamine (lis) and dextroamphetamine (d-amp) on $BK^{WT}$ and $BK^{N999S}$ channels expressed in HEK293 cells.

**Figure supplement 2—source data 1.** Data file for *Figure 1—figure supplement 2*.

described as dystonia, hypotonia, non-narcoleptic cataplexy, and akinesia. PNKD3 episodes manifest as postural lapses with immobility ('drop attacks') that occur with short duration and high frequency, often hundreds of times per day (*Du et al., 2005*; *Heim et al., 2020*; *Keros et al., 2022*; *Li et al., 2018*; *Wang et al., 2017*; *Zhang et al., 2015*). Just under half of patients experience seizure of varying types, including absence, atonic, myoclonic, and generalized tonic-clonic (GTC). However, epilepsy and PNKD are not consistently co-morbid (*Du et al., 2005*; *Miller et al., 2021*). Individuals with putative LOF variants report additional movement disorders including dyskinesia, axial hypotonia, tremor, or ataxia, in addition to various seizure types (*Du et al., 2020*; *Liang et al., 2019*; *Rodrigues Bento et al., 2021*; *Tabarki et al., 2016*; *Yeşil et al., 2018*). It is not yet clear whether variations in symptomatic presentation result from incomplete or inconsistent clinical evaluations, or genuine genotype-phenotype differences within either GOF or LOF cohorts.

We address these questions through heterologous, neuronal, and neurobehavioral validation for three patient-associated *KCNMA1* variants in mouse models. The GOF $BK^{N999S}$ and $BK^{D434G}$ channels produced increased neuronal BK currents and firing as heterozygous alleles in transgenic mice, while heterozygous LOF $BK^{H444Q}$ channels were insufficient to alter neuronal properties. Mice were evaluated in a series of spontaneous and evoked seizure and locomotor assays. N999S propagated the largest symptomatic burden with chemoconvulsant challenge and stress-triggered dyskinesia, supporting the conclusion that this variant has the greatest monogenic pathogenicity, followed by D434G, $Kcnma1^{-/-}$, and H444Q. The results identify $Kcnma1^{N999S/WT}$ mice as a PNKD3 model with the highest phenotypic similarity to patients harboring *KCNMA1* GOF variants, including symptomatic resolution with acute dextroamphetamine treatment. Our findings further establish the stress-induced PNKD assay to delineate distinct symptomatic manifestations between GOF and LOF alleles, supporting its utility in a battery of neurobehavioral evaluations to define *KCNMA1*-linked channelopathy models.

## Results

### Patient variants confer GOF (N999S and D434G) and LOF (H444Q) properties on BK channel activity

A comparative assessment for three dyskinesia-associated patient variants (N999S, D434G, and H444Q) was performed within the human BK channel (*Figure 1*). BK channel function was assessed using inside-out patch-clamp recordings in HEK293 cells. Patches from cells expressing $BK^{WT}$, $BK^{N999S}$, $BK^{D434G}$, $BK^{H444Q}$ channels were activated with depolarizing voltage steps, and the voltage dependence of activation and kinetics were assessed from macroscopic currents (*Figure 1C*). Conductance versus voltage (G-V) relationships (*Figure 1D*) were assessed by the voltage of half-maximal activation ($V_{1/2}$; *Figure 1E*).

$BK^{WT}$ currents had a $V_{1/2}$ of 125±2 mV. Introduction of N999S and D434G mutations shifted the G-V relationships to more negative membrane potentials ($V_{1/2}$: $BK^{N999S}$ 41±3 mV and $BK^{D434G}$ 88±2 mV), confirming their GOF effect at all voltages. The decrease in $V_{1/2}$ for $BK^{N999S}$ channels compared to $BK^{WT}$ was 20–30 mV larger than observed in prior studies with different splice variants and intracellular $Ca^{2+}$ (*Li et al., 2018*; *Moldenhauer et al., 2020a*). Here, under equivalent conditions, N999S produced a larger hyperpolarizing shift from WT ($\Delta V_{1/2}$ = 84 mV) versus D434G ($\Delta V_{1/2}$ = 37 mV). In addition, N999S and D434G produced faster BK channel activation and slower deactivation compared to WT (*Figure 1F–G*). Altogether, $BK^{N999S}$ channels showed greater GOF properties than $BK^{D434G}$ in all parameters under these conditions, corroborating the relative severity predicted from prior work.

In contrast, introduction of the H444Q variant shifted the G-V relationship to more positive potentials ($V_{1/2}$: $BK^{H444Q}$ 155±4 mV), consistent with LOF effects. H444Q produced changes in channel opening and closing further consistent with LOF effects, slowing activation and speeding deactivation (*Figure 1D–G*). H444Q produced a smaller difference from WT than either GOF variant ($\Delta V_{1/2}$ = 30 mV), identifying H444Q as a comparatively mild variant. The results indicate that N999S produces the strongest effect on BK channel activation in the GOF direction, followed by D434G (GOF) and H444Q (LOF).

### Generation of N999S, D434G, and H444Q mouse models

Correlation between patient genotype and phenotype has only been established for a single *KCNMA1* variant so far, D434G, an autosomal dominant that co-segregates with PNKD and epilepsy in a multi-generation pedigree (*Du et al., 2005*). D434G pathogenicity is further corroborated by mouse and fly models, which show alterations in neuronal excitability, brain and motor function (*Dong et al., 2021*; *Kratschmer et al., 2021*). In contrast, N999S and H444Q lack this direct evidence due to the absence of familial transmission among the children that carry these variants (*Miller et al., 2021*). N999S is the most common de novo KCNMA1 variant (~17% of all patients), found as heterozygous in every case. About half of individuals harboring N999S alleles are diagnosed with seizure, PNKD, or both (*Keros et al., 2022*; *Li et al., 2018*; *Wang et al., 2017*; *Zhang et al., 2015*), suggesting a strong potential to be causative in channelopathy symptoms. H444Q is found in a single case and is one of several putative LOF variants where affected individuals have dyskinesia-like paroxysms (*Miller et al., 2021*). This proband had a history of abnormal EEG, unresolved with respect to the diagnosis of epilepsy, but also harbors three additional genetic findings (ALM unpublished data).

To establish genotype-phenotype correlations, heterozygous mice replicating the patient genotypes were first evaluated. Each variant was introduced as a single nucleotide mutation into the mouse *Kcnma1* gene using CRISPR base-editing (*Figure 1—figure supplement 1*). In all animal experiments, investigators were blinded to genotype during data collection and analysis, and WT controls were compared to transgenic littermates within individual transgenic lines. *Kcnma1*^N999S/WT^, *Kcnma1*^D434G/WT^, and *Kcnma1*^H444Q/WT^ mice were grossly behaviorally and morphologically normal with no notable spontaneous paroxysms, gait abnormalities, or visually detectable seizures during home cage observation. *Kcnma1*^D434G/WT^ and *Kcnma1*^H444Q/WT^ intercrosses produced homozygous progeny that were also visually normal. However, *Kcnma1*^N999S/WT^ intercrosses produced no homozygous pups (see Materials and methods). The absence of homozygous N999S progeny is similar to Tg-BK^R207Q^ mice harboring another strong GOF mutation that showed lethality in the homozygous allele configuration (*Montgomery and Meredith, 2012*). Given the inability to generate homozygous N999S mice, gene expression was analyzed from hippocampus and cerebellum of *Kcnma1*^N999S/WT^ and WT littermates (n=3 mice each genotype and tissue). No significant differences were found in the levels of *Kcnma1* (1.07-fold change,

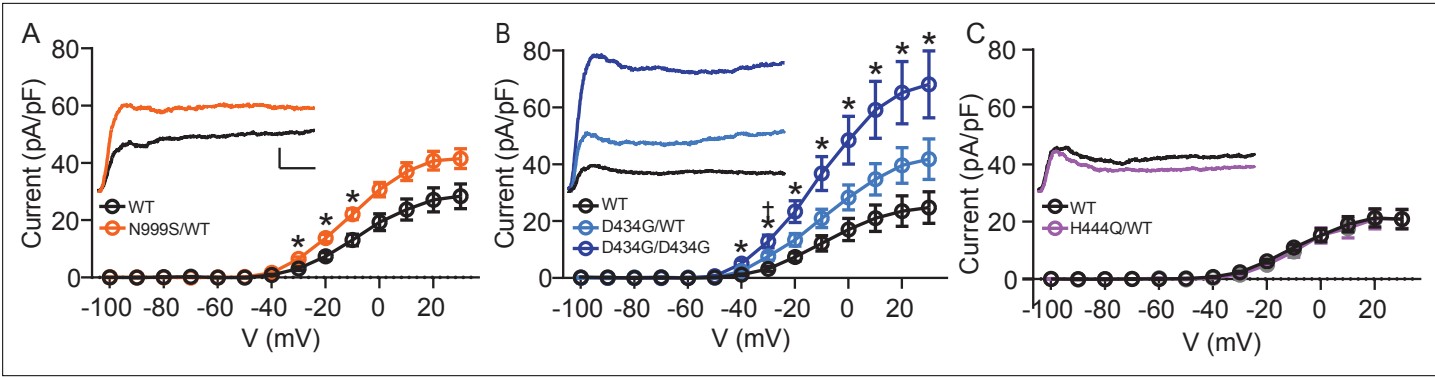

**Figure 2.** Increased BK current in *Kcnma1*[N999S/WT] and *Kcnma1*[D434G/D434G] granule neurons. Whole-cell macroscopic BK currents were recorded in 1 µM tetrodotoxin (TTX) and 2 mM 4-aminopyridine (4-AP), isolated with 10 µM paxilline, and normalized to cell capacitance. Activating voltage steps were applied from $V_h$ of −90 mV, stepping from −100 to +30 mV for 150 ms, and back to −90 mV for 130 ms. (A−C) Peak BK current density versus voltage relationships. Data are presented as mean ± SEM. * and †, p<0.05, two-way repeated measures ANOVA with Bonferroni post hoc. *Insets:* Representative BK current traces at 30 mV. Scale bars: 500 pA, 5 ms. (**A**) BK current density was larger in *Kcnma1*[N999S/WT] neurons (n=16 neurons, 5 mice) compared to *Kcnma1*[WT/WT] (n=14 neurons, 4 mice) at −30 mV (p=0.0114), −20 (p=0.0210), −10 (p=0.0426) voltage steps (indicated with *). (**B**) BK current density was larger in *Kcnma1*[D434G/D434G] neurons (n=12 neurons, 3 mice) compared to *Kcnma1*[WT/WT] (n=10 neurons, 4 mice) at density at −40 mV (p=0.0112), −30 (p=0.0026), −20 (p=0.0031), −10 (p=0.0038), 0 (p=0.0078), 10 (p=0.0068), 20 (p=0.0071), 30 (p=0.0088) voltage steps (*). *Kcnma1*[D434G/WT] mice (n=9 neurons, 3 mice) had higher BK current density compared to *Kcnma1*[WT/WT] at −30 mV only (†p=0.0321). (**C**) BK current density was not different in *Kcnma1*[H444Q/WT] neurons (n=7 neurons, 2 mice) compared to *Kcnma1*[WT/WT] (n=6 neurons, 3 mice).

The online version of this article includes the following source data for figure 2:

**Source data 1.** Data file for *Figure 2A–C*.

p=0.79, FDR = 0.99, ANOVA with eBayes test), or BKβ subunits expressed in brain: *Kcnmb1* (−1.07, p=0.14), *Kcnmb2* (1.00, p=0.84), and *Kcnmb4* (−1.03, p=0.46). The results show no gross up- or down-regulation of BK channel components in either the hippocampus or cerebellum of *Kcnma1*[N999S/WT] neurons. However, since the microarray-based probe set does not distinguish between the WT and N999S *Kcnma1* transcripts, no further conclusion can be made regarding the specific expression ratio of each allele.

## N999S and D434G increase BK current in hippocampal neurons

From heterologous cells, we predicted that the variants would have a strong (N999S), intermediate (D434G), or weak (H444Q) potential to alter neuronal BK current levels in transgenic mice. However, heterozygous patient genotypes create the possibility for hetero-tetramer channel formation (*Geng et al., 2021*), necessitating understanding the relative GOF and LOF effects in vivo from BK current levels in heterozygous transgenics compared to WT littermates. Recordings were made in the dentate gyrus of the hippocampus, where BK channels are highly expressed, regulate neuronal excitability, and where changes in BK channel properties are associated with seizure (*Kaufmann et al., 2010*; *Knaus et al., 1996*; *Misonou et al., 2006*; *Sailer et al., 2006*; *Sausbier et al., 2005*; *Sausbier et al., 2006*; *Trimmer, 2015*). In dentate granule cells, excitability is sensitive to changes in BK current in both directions, assessed using pharmacological inhibition as well as deletion of the β4 subunit (*Brenner et al., 2005*; *Mehranfard et al., 2014*; *Mehranfard et al., 2015*). Loss of β4 creates GOF BK channels by speeding BK channel kinetics, and *Kcnmb4*[−/−] knockout mice have temporal lobe seizures (*Jaffe and Brenner, 2018*; *Petrik et al., 2011*; *Wang et al., 2016*; *Whitmire et al., 2017*).

BK currents from *Kcnma1*[WT/WT] neurons activated at −40 mV, increasing to 21–28 pA/pF at the highest voltage across mouse strains (*Figure 2A–C*). *Kcnma1*[N999S/WT] neurons had a 69% increase in BK current compared to WT littermates (*Kcnma1*[WT/WT] 13.0±2.0 pA/pF and *Kcnma1*[N999S/WT] 22.0±1.8 pA/pF at −10 mV; *Figure 2A*). The increased BK current likely results from alterations in BK channel activity, since *KCNMA1* expression was not changed in *Kcnma1*[N999S/WT] neurons. *Kcnma1*[D434G/WT] BK currents were 73% larger (*Kcnma1*[WT/WT] 12.1±2.6 pA/pF and *Kcnma1*[D434G/WT] 20.9±3.2 pA/pF at −10 mV; *Figure 2B*), although not statistically different at most voltages due to variability. However, two copies of the D434G variant (*Kcnma1*[D434G/D434G]) resulted in the largest increase in BK current across voltages from −40 mV to the maximum (203%; 36.7±5.9 pA/pF −10 mV). Interestingly, by direct

comparison $Kcnma1^{D434G/WT}$ BK current levels were similar to $Kcnma1^{N999S/WT}$, despite the more severe phenotype for BK$^{N999S}$ channels in heterologous cells (*Figure 1*).

In contrast, BK currents in $Kcnma1^{H444Q/WT}$ neurons were not significantly different compared to WT littermates at any voltage ($Kcnma1^{WT/WT}$ 10.9±1.0 pA/pF and $Kcnma1^{H444Q/WT}$ 9.5±1.9 pA/pF at −10 mV; −13% change; *Figure 2C*). This establishes an allelic series of $Kcnma1^{D434G/D434G} >> Kcnma1^{D434G/WT} ≈ Kcnma1^{N999S/WT} > Kcnma1^{H444Q/WT}$ with respect to BK current magnitude and supports the potential for N999S and D434G to cause neurobehavioral changes. The detrimental potential for $Kcnma1^{H444Q/WT}$ is less clear and may require additional factors or mechanisms to support pathogenicity (i.e., other Ca$^{2+}$ conditions, cell types, or gene interactions).

These dentate granule whole-cell recordings represent an initial evaluation of the potential for each variant to affect BK currents under physiological conditions. For $Kcnma1^{N999S/WT}$, the increased BK current is not associated with higher transcript expression. Although the ratio of WT and N999S transcripts could not be individually determined from the microarray probe set, the single nucleotide mutations introduced into coding exons provide no obvious mechanism to alter the allelic expression ratios. Assuming both alleles are expressed normally, it would suggest functional changes underlie the BK current increase. However, how the properties identified from homotetramic channels (*Figure 1C–G*) contribute to the increased current in neurons is unknown. The apparent voltage dependence of activation was not different between N999S, D434G, H444Q, and their respective WT control currents in neurons (data not shown). Several factors that could mitigate differences in V$_{1/2}$ recorded from homotetramers are undefined in the neurons, including the α (WT:mutant) and β subunit stoichiometry, splice variant background, and intracellular Ca$^{2+}$. Limited data is available to consider the impact of these on heterozygous variants. Co-expression of WT and mutant (GOF) BK channel cDNAs supports the assumption that heterotetramers are the predominant channel type produced by 1:1 transcript ratios in *Xenopus* oocytes (*Geng et al., 2021*). A few studies have shown that N999S and D434G confer similar ΔV$_{1/2}$ onto different splice variants (*Figure 1C–G*; *Li et al., 2018*; *Moldenhauer et al., 2020a*; *Wang et al., 2009*) and maintain left-shifted V$_{1/2}$ values compared to WT in the presence of the β4 subunit (*Berkefeld and Fakler, 2013*; *Li et al., 2018*; *Wang et al., 2009*). This data is not available for H444Q, which produced smaller effects. Yet even with D434G, less of a difference is found ±β4 above 10 μM Ca$^{2+}$ (*Wang et al., 2009*), which could be significant in granule neurons given the widespread abundance of β4.

## N999S and D434G increase intrinsic neuronal excitability

Intrinsic excitability was next assessed in dentate granule neurons as an independent validation for neuronal pathogenicity. Both GOF and LOF BK channel mutations have the ability to alter neuronal activity in either direction, depending on the context (*Bailey et al., 2019*; *Brenner et al., 2005*; *Montgomery and Meredith, 2012*; *Gu et al., 2007*; *Sausbier et al., 2004*). Dentate granule cell input-output firing relationships were assessed in current-clamp mode (*Figure 3*). Firing rates increased with current injection in each $Kcnma1^{WT/WT}$ littermate control dataset, reaching a peak of ~40 Hz between 240 and 260 pA and then decreasing with higher current injections (*Figure 3A–B*). $Kcnma1^{N999S/WT}$ firing was greater than $Kcnma1^{WT/WT}$ littermate neurons in several key places. First, across the whole current injection range, firing was significantly increased in the middle portion (160–240 pA), ranging from 25% to 30% higher than WT (*Figure 3Ai*, Bi). After reaching the maximum, the firing still decreased instead of remaining higher through the full range of current injections. In addition, the initial slope of firing (0–160 pA) was greater in $Kcnma1^{N999S/WT}$ neurons (0.22±0.01 Hz/pA) compared to $Kcnma1^{WT/WT}$ (0.18±0.01 Hz/pA, *Figure 3Ci*). Lastly, the maximal firing was 9.6±1.8 Hz (125%) higher in $Kcnma1^{N999S/WT}$ neurons versus $Kcnma1^{WT/WT}$ (*Figure 3Di*). Taken together, $Kcnma1^{N999S/WT}$ neurons respond to stimulation with higher firing and a shift in the input-output relationship.

Increased firing was also observed in $Kcnma1^{D434G/WT}$ neurons, but the shape of the input-output alteration was different than that observed in $Kcnma1^{N999S/WT}$. Firing was 18–67% greater than WT controls at higher current injections only, from 260 to 400 pA (*Figure 3Aii,Bii*). Despite the increase at the higher end of the range, $Kcnma1^{D434G/WT}$ firing still decreased after reaching a maximum, while remaining higher than $Kcnma1^{WT/WT}$. The initial firing rate slope was not different from $Kcnma1^{WT/WT}$ (*Figure 3Cii*). However, the maximal firing rate was 5.5±2.2 Hz greater (113%) for $Kcnma1^{D434G/WT}$ compared to $Kcnma1^{WT/WT}$ (*Figure 3Dii*). This increase was shifted to higher current injections and occurred over a wider range of voltages than that observed for $Kcnma1^{N999S/WT}$.

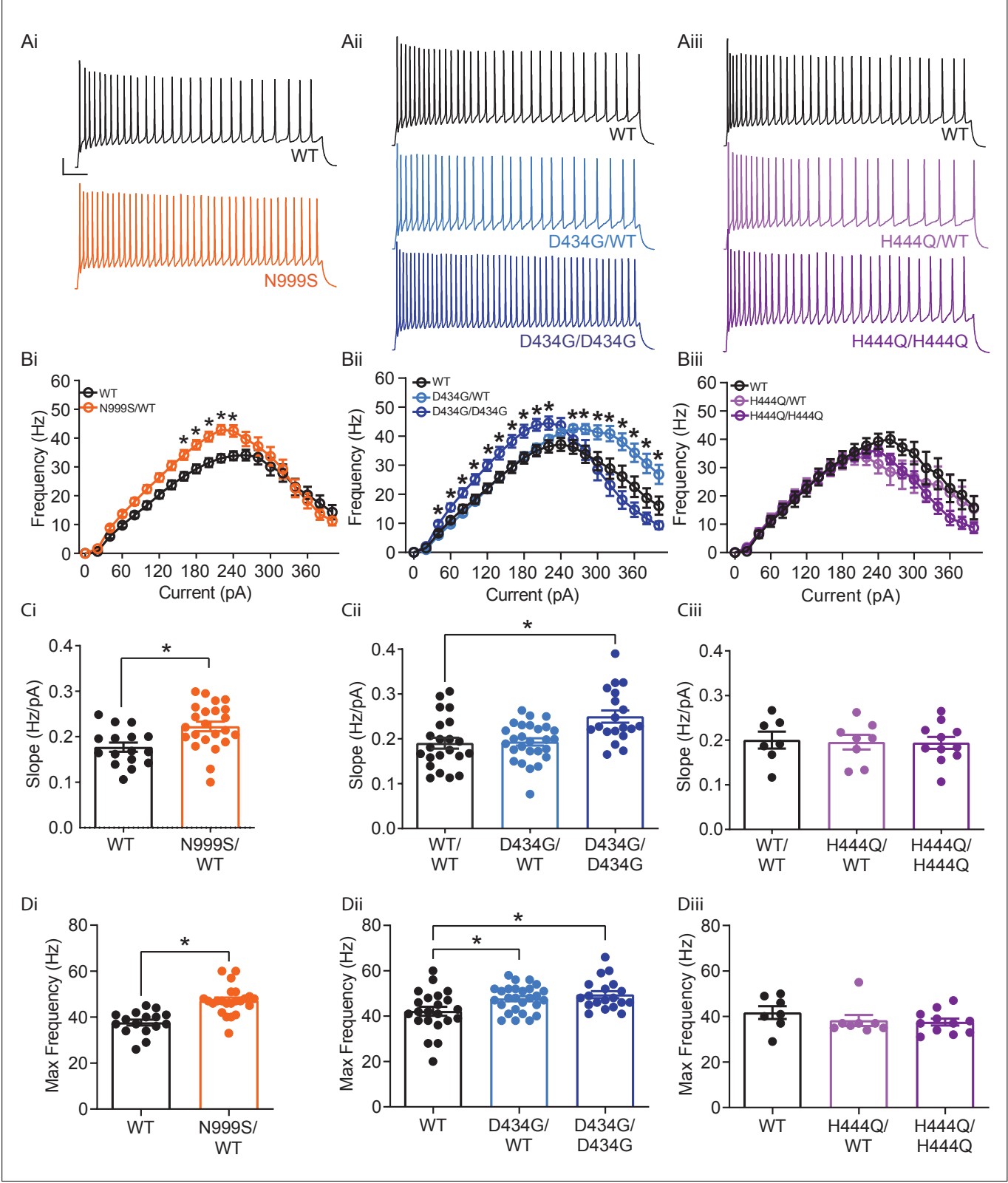

**Figure 3.** Increased intrinsic excitability in *Kcnma1*[N999S/WT], *Kcnma1*[D434G/WT], and *Kcnma1*[D434G/D434G] granule neurons. In current-clamp mode, step currents from 0 to 400 pA were applied to dentate granule neurons under the same ionic conditions used to record BK currents. (Ai–Aiii) Representative AP trains elicited from the 200 pA current injection step in WT and transgenic neurons. Scale bar: 20 mV, 100 ms. (Bi–Biii) Input-output relationship for firing frequency versus step current injection. Data are presented as mean ± SEM. *p<0.05, two-way repeated measures ANOVA with Bonferroni post

*Figure 3 continued on next page*

*Figure 3 continued*

hoc. (Bi) *Kcnma1*[N999S/WT] (n=23 neurons, 5 mice) firing was higher than *Kcnma1*[WT/WT] (n=16 neurons, 5 mice) at 160 pA (p=0.0426), 180 pA (p=0.0143), 200 (p=0.0068), 220 pA (p=0.0009), and 240 pA (p=0.0337) current steps. (Bii) *Kcnma1*[D434G/WT] (n=27 neurons, 5 mice) firing was higher than *Kcnma1*[WT/WT] (n=22 neurons, 5 mice) at 260 pA (p=0.0452), 280 (p=0.0314), 300 (p=0.0351), 320 (p=0.0177), 340 (p=0.0309), 360 (p=0.0358), 380 (p=0.0312), and 400 (p=0.0444) current steps. *Kcnma1*[D434G/D434G] (n=19 neurons, 4 mice) firing was higher than *Kcnma1*[WT/WT] at 40 pA (*p*=0.0266), 60 (p=0.0233), 80 (p=0.0277), 100 (p=0.0130), 120 (p=0.0074), 140 (p=0.0119), 160 (p=0.0084), 180 (p=0.0063), 200 (p=0.0059), and 220 (p=0.0261) current steps. (Biii) *Kcnma1*[H444Q/WT] (n=8 neurons, 2 mice) and *Kcnma1*[H444Q/H444Q] (n=11 neurons, 2 mice) firing was not different than *Kcnma1*[WT/WT] (n=7 neurons, 1 mouse) at any current step (p=0.3222). (Ci–Ciii) Initial slope for the firing rate gain between 0 and 160 pA current injections. Data are presented as mean ± SEM, with individual data points. (Ci) *Kcnma1*[N999S/WT] firing slope was increased compared to WT (*p=0.0034; t-test). (Cii) *Kcnma1*[D434G/D434G] firing slope was increased compared to WT (*p=0.0051; one-way ANOVA), *Kcnma1*[D434G/WT] slopes were unchanged (p=0.9774). (Ciii) *Kcnma1*[H444Q/WT] and/or *Kcnma1*[H444Q/H444Q] firing slopes were not different than WT (p=0.9658). (Di–Diii) Maximum firing frequency. Data are presented as mean ± SEM. (Di) Maximal firing from *Kcnma1*[N999S/WT] neurons was increased compared to WT (*p<0.0001; t-test). (Dii) Maximal firing from *Kcnma1*[D434G/WT] and *Kcnma1*[D434G/D434G] neurons was increased compared to WT (*p=0.0387 and p=0.0111, respectively; one-way ANOVA). (Diii) Maximal firing from *Kcnma1*[H444Q/WT] and/or *Kcnma1*[H444Q/H444Q] neurons was not different than WT (p=0.4625; one-way ANOVA). Passive membrane properties for this dataset appear in *Figure 3—figure supplement 1*. Action potential waveform analysis for this dataset appears in *Figure 3—figure supplement 2*.

The online version of this article includes the following source data and figure supplement(s) for figure 3:

**Source data 1.** Data file for *Figure 3B–D*.

**Figure supplement 1.** Passive membrane properties.

**Figure supplement 1—source data 1.** Data file for *Figure 3—figure supplement 1*.

**Figure supplement 2.** *Kcnma1*[N999S/WT], *Kcnma1*[D434G/WT], and *Kcnma1*[D434G/D434G] action potential waveforms.

**Figure supplement 2—source data 1.** Data file for *Figure 3—figure supplement 2B-H*.

*Kcnma1*[D434G/D434G] neurons, which had the highest BK current levels, showed further differences from *Kcnma1*[D434G/WT]. Firing was increased 22–47% in the early and middle of the current injection range, from 40 to 220 pA (*Figure 3Aii, Bii*). Both the initial slope (*Kcnma1*[D434G/D434G] 0.25±0.01 Hz/pA versus 0.19±0.01 Hz/pA for *Kcnma1*[WT/WT]) and the maximal firing rate were greater (117%, *Figure 3Cii, Dii*). Yet *Kcnma1*[D434G/D434G] firing was qualitatively more similar to *Kcnma1*[N999S/WT], despite the finding that *Kcnma1*[D434G/D434G] BK current levels were almost twice as much as those recorded from *Kcnma1*[N999S/WT].

No significant differences in firing frequency, slope for the initial firing rate gain, or maximal firing rate were observed in *Kcnma1*[H444Q/WT] or *Kcnma1*[H444Q/H444Q] neurons compared to WT littermates (*Figure 3Aiii–Diii*). The lack of significant alteration in excitability was congruent with the absence of change in BK current levels in *Kcnma1*[H444Q/WT] neurons.

We conclude that both the N999S and D434G GOF variants have pathogenic potential through their ability to increase BK currents and action potential firing. The LOF H444Q variant does not substantiate the same pathogenic potential under these conditions. Mechanistically, despite the grossly similar BK current levels between *Kcnma1*[D434G/WT] and *Kcnma1*[N999S/WT] neurons, the non-identical input-output curves suggest a more complex relationship between BK channel properties and neuronal excitability in dentate granule neurons. Hints about the basis for these differences may be revealed by comparison of additional membrane parameters such as passive membrane properties and action potential waveforms. However, no differences in resting membrane potential or input resistance were observed between *Kcnma1*[N999S/WT], *Kcnma1*[D434G/WT], or *Kcnma1*[D434G/D434G] and WT controls (*Figure 3—figure supplement 1*). With respect to action potentials, modulation of repolarization and the afterhyperpolarization (AHP) in repetitive firing occurs with both BK channel inhibition (slower repolarization and reduced AHP amplitude) and activation (faster repolarization and increased AHP amplitude) (*Dong et al., 2021*; *Gu et al., 2007*; *Montgomery and Meredith, 2012*; *Shao et al., 1999*). Analysis of waveforms from the 200 pA step corroborate BK channels regulate multiple phases of the action potential, but suggested that the basis for increased firing in *Kcnma1*[N999S/WT] and *Kcnma1*[D434G/D434G] neurons was a faster AHP decay rate (*Figure 3—figure supplement 2*), which would facilitate more rapid initiation of the next action potential. Since *Kcnma1*[D434G/WT] neurons do not show increased firing at 200 pA, no differences would be expected in parameters related to setting firing frequency. Accordingly, no significant difference in AHP decay was found in *Kcnma1*[D434G/WT] versus control waveforms.

The mechanism by which GOF BK channels facilitate AHP decay is not revealed in this study. Though the N999S and D434G variants both slow deactivation in heterologous cells, the deactivation rate remains to be defined in *Kcnma1*[N999S/WT] and *Kcnma1*[D434G/D434G] neurons under repetitive firing

conditions with dynamic Ca$^{2+}$. In dentate gyrus, the GOF variants share some similarity to BK currents and action potential waveforms recorded from neurons lacking the β4 subunit (*Brenner et al., 2005*; *Jaffe and Brenner, 2018*; *Wang et al., 2016*). *Kcnmb4$^{-/-}$* neurons have increased BK current and increased firing associated with accelerated an AHP decay. Like *Kcnmb4$^{-/-}$*, GOF BK currents may speed AHP decay rate indirectly by affecting another current, most likely SK current due to the lack of change in the AHP amplitudes. GOF variants in the context of β4 deletion would be predicted more severe than either alone, potentially speeding repolarization and further shortening the AHP.

## N999S and D434G reduce seizure thresholds in mice

Neuronal hyperexcitability is coincident with establishment of an epileptic network, and about half of all individuals with *KCNMA1* channelopathy, including those with N999S, D434G, and H444Q variants, report a history of seizures or epilepsy (*Bailey et al., 2019*; *Miller et al., 2021*). Individuals harboring the D434G variant primarily have absence seizures, if present (*Du et al., 2005*). Dentate gyrus hyperexcitability can both contribute to, and result from, epileptiform activity (*Dengler et al., 2017*; *Krook-Magnuson et al., 2015*; *Mehranfard et al., 2015*; *Scharfman, 2019*). In β4$^{-/-}$ mice, increased granule neuron firing is found in the setting of hippocampal epileptiform discharges, non-convulsive seizures, and lower chemoconvulsant-induced seizure thresholds (*Brenner et al., 2005*; *Whitmire et al., 2017*). We hypothesized that *Kcnma1$^{N999S/WT}$* and *Kcnma1$^{D434G/WT}$* mice would show increased number, duration, or severity of seizure events compared to WT controls. However, since half of those harboring LOF variants also report seizures (*Liang et al., 2019*; *Miller et al., 2021*), including the H444Q and individuals with putative truncation alleles, *Kcnma1$^{H444Q/WT}$* and *Kcnma1$^{-/-}$* mice were assessed in parallel. No seizures have been previously reported in two established *Kcnma1$^{-/-}$* mouse models (*Bailey et al., 2019*; ALM unpublished data), but spontaneous epilepsy was reported in a *Kcnma1*-exon4 frameshift mouse line (*Yao et al., 2021*).

Behavioral assessments and EEGs were made from transgenic and WT littermates for indications of seizure. No spontaneous twitching/jumping/convulsions, rigidity/immobility, anorexia/dehydration, or premature mortality were observed from transgenic (or control) mice in the home cage environment. After dural electrode implantation, 24 hr baseline EEGs were recorded. No interictal epileptiform discharges, spontaneous seizures, or other abnormalities (e.g., slowing) were observed in transgenic or control mice during baselines. The absence of spontaneous events was not surprising given that half of affected individuals do not report epilepsy, and among those that do, there is a wide range in frequency (isolated to daily), semiology, and age of onset (*Bailey et al., 2019*; *Miller et al., 2021*). However, this presents challenges to evaluating spontaneous EEG events in mouse models, especially those that could be occurring in deeper brain regions similar to β4$^{-/-}$ mice. The presence of EEG abnormalities could be more comprehensively assessed with longer monitoring, depth electrodes, or interrogation of additional ages and strain backgrounds (*Löscher et al., 2017*), which were beyond the capability of the present study.

Human epilepsy variants in rodent models without spontaneous abnormalities often exhibit decreased thresholds to triggered seizures (*Feliciano et al., 2011*; *Watanabe et al., 2000*; *Yuskaitis et al., 2018*), although this is not entirely predictive of epilepsy risk (*Noebels, 2003*). We hypothesized that *Kcnma1$^{N999S/WT}$* and *Kcnma1$^{D434G/WT}$* mice would show either decreased threshold or increased severity with 40 mg/kg pentylenetetrazol (PTZ) chemoconvulsant challenge. *Kcnma1$^{WT/WT}$* controls for each line developed seizures consistent with those observed with PTZ in other studies (*Van Erum et al., 2019*) ranging from abnormal posturing and myoclonic twitching (10/18 mice; modified Racine score 1 or 2) to tonic-clonic activity (7/18 mice; modified Racine 3 or 4) within minutes after PTZ injection (*Figure 4—video 4–1*).

*Kcnma1$^{N999S/WT}$* mice developed PTZ-induced seizures that were distinguishable from *Kcnma1$^{WT/WT}$* littermates in several parameters. Behaviorally, most *Kcnma1$^{N999S/WT}$* mice displayed tonic-clonic activity (9/13 mice modified Racine 3 or 4), with two reaching status epilepticus (2/13 mice; modified Racine 5). The latency to first seizure after PTZ injection was reduced to 75±15 s, compared to WT littermates (294±99 s; *Figure 4Ai,C-D*). EEG power, an estimation of seizure severity, showed a broader range with *Kcnma1$^{N999S/WT}$* mice, although the differences were not significant (*Figure 4Bi,C-D*). Interestingly despite these observations, mice exhibiting electrographic seizures did not look strikingly behaviorally different from control mice. One reason may be the movement suppression that developed in *Kcnma1$^{N999S/WT}$* mice after PTZ injection, quantified by EMG. After PTZ, *Kcnma1$^{WT/}$*

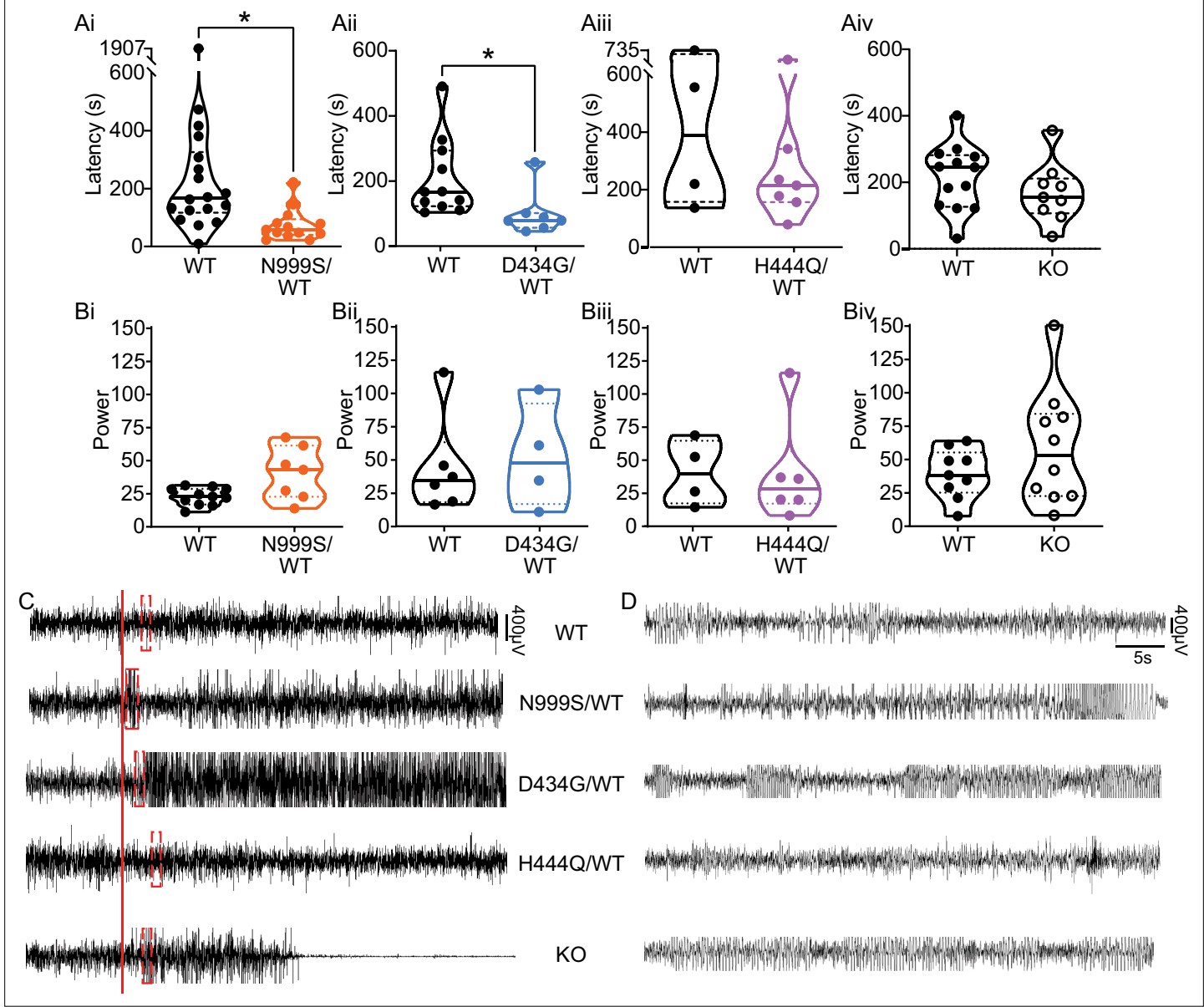

**Figure 4.** Pentylenetetrazol (PTZ)-induced seizures in mice. (Ai–Aiv) Latency to initial seizure after PTZ injection. Data are individual mice with median and inter-quartile range. (Ai) Latency was decreased in *Kcnma1*[N999S/WT] mice (n=13) compared to *Kcnma1*[WT/WT] (n=18, *p=0.0006; Mann-Whitney test). (Aii) Latency was decreased in *Kcnma1*[D434G/WT] mice (n=7) compared to *Kcnma1*[WT/WT] (n=11, *p=0.0041; Mann-Whitney test). (Aiii) Seizure latency was comparable between *Kcnma1*[H444Q/WT] (n=7) and *Kcnma1*[WT/WT] (n=4, p=0.5273; Mann-Whitney test). (Aiv) No differences were found in seizure latency between *Kcnma1*[−/−] (n=9) and *Kcnma1*[+/+] mice (n=13, p=0.2282; Mann-Whitney test). (Bi–iv) Total EEG power after PTZ injection (y-axis in $\mu V^2/Hz \times 10^2$). Data are individual mice with median and inter-quartile range. (Bi) EEG power was not different between *Kcnma1*[N999S/WT] (n=7) and *Kcnma1*[WT/WT] (n=11, p=0.0619; t-test). (Bii) *Kcnma1*[D434G/WT] (n=4) was not different from *Kcnma1*[WT/WT] (n=6, p=0.7563; t-test). (Biii) *Kcnma1*[H444Q/WT] (n=6) was not different from *Kcnma1*[WT/WT] (n=4, p=0.9641; t-test). (Biv) *Kcnma1*[−/−] (n=10) was not different from *Kcnma1*[+/+] (n=9, p=0.2134; t-test). (C) Representative EEG traces over 45 min at baseline and after PTZ injection (red line). (D) Expanded EEG traces for the first seizure indicated with the red boxes in (C). Representative videos for this dataset appear in *Figure 4—videos 1–4*.

The online version of this article includes the following video and source data for figure 4:

**Source data 1.** Data file for *Figure 4A, B*.

**Figure 4—video 1.** *Kcnma1*[WT/WT].

https://elifesciences.org/articles/77953/figures#fig4video1

**Figure 4—video 2.** *Kcnma1*[N999S/WT].

https://elifesciences.org/articles/77953/figures#fig4video2

*Figure 4 continued on next page*

*Figure 4 continued*

**Figure 4—video 3.** *Kcnma1*[D434G/WT].
https://elifesciences.org/articles/77953/figures#fig4video3

**Figure 4—video 4.** *Kcnma1*[−/−].
https://elifesciences.org/articles/77953/figures#fig4video4

[WT] mice had infrequent bouts of sustained quiescent EMG activity, with average lengths of 45±7 s (n=16). However, the inactive bouts were longer for *Kcnma1*[N999S/WT] mice (311±126, n=10, p<0.0001, Mann-Whitney test) and were visually apparent (*Figure 4—video 4–2*). The movement suppression exhibited by *Kcnma1*[N999S/WT] mice under PTZ does not have a correlate in individuals harboring N999S variants, although a few report absence seizures among other types (*Miller et al., 2021*). Since no spontaneous EEG[+]/EMG[−] events were observed in the baseline EEG recording period of these mice, it remains to be determined whether the PTZ-elicited movement suppression is related to an absence-like seizure manifestation.

Within the D434G family, there is an intermediate penetrance for epilepsy (56%), the most frequent diagnosis being absence (*Du et al., 2005*; *Miller et al., 2021*). Like N999S, *Kcnma1*[D434G/WT] mice also showed a reduced latency to first seizure (101±27 s) compared to *Kcnma1*[WT/WT] mice (209±35 s; *Figure 4Aii,C-D*). However, this reduction was not as large as the difference between *Kcnma1*[N999S/WT] mice and their respective controls. Total EEG power from *Kcnma1*[D434G/WT] mice was not different from WT controls (*Figure 4Bii*). Therefore, the D434G variant also increased the propensity for seizure in the transgenic model, consistent with its ability to alter neuronal excitability, but was less severe than the N999S variant. The phenotype assessed here for *Kcnma1*[D434G/WT] is also less severe than reported in a knock-in mouse model with the D434G mutation introduced in the context of a Cre/lox cassette. Those mice showed spontaneous spike-wave discharges in both the heterozygous and homozygous configuration with complete penetrance (*Dong et al., 2021*), a phenotype that appears more severe than reported in the D434G pedigree, in which only half experience seizures (*Du et al., 2005*).

Although no patients have a homozygous D434G genotype, a limited number of *Kcnma1*[D434G/D434G] mice were available for EEG analysis. We tested whether *Kcnma1*[D434G/D434G] mice, producing only mutant BK channel homotetramers, had a more severe phenotype. These mice showed a trend toward the shortest latencies to seizure, with thresholds comparable to the lowest among the *Kcnma1*[D434G/WT] cohort (48, 58, and 69 s). *Kcnma1*[D434G/D434G] mice had EEG power overlapping with WT and heterozygous littermates (2172, 4237, and 3627). While not conclusive and requiring substantiation in a larger cohort, the *Kcnma1*[D434G/D434G] seizure data raise the possibility of a gene dosage effect with D434G that qualitatively differs from an independently generated D434G mouse model (*Dong et al., 2021*).

For LOF transgenics, changes in seizure threshold in both directions were considered. Approximately the same proportion of individuals with LOF variants report seizure as those with GOF variants (*Miller et al., 2021*), a finding validated in one *Kcnma1*[−/−] mouse model (*Yao et al., 2021*). Yet paradoxically, acute inhibition of BK channels has anti-seizure effects in other rodent models (*Dong et al., 2021*; *Kuebler et al., 2001*; *Sheehan et al., 2009*). However, no differences were observed in latency to first seizure or total EEG power in *Kcnma1*[H444Q/WT] or *Kcnma1*[−/−] mice (*Figure 4Aiii-Aiv,Biii-Biv, C-D*; *Figure 4—video 4*). For H444Q, this data suggests the lack of change in dentate granule neuron BK currents and excitability may be consistent with other areas of the brain, producing no change in seizure propensity indicative of widespread hyperexcitability. Overall, alterations in BK current and firing detected concurrently with lowered seizure threshold suggests that the GOF variants N999S and D434G have the potential to contribute to seizure risk by changing neuronal activity in a mouse model. However, partial (H444Q) or total (*Kcnma1*[−/−]) loss of BK channel function does not support the same potential in seizure etiology under equivalent conditions.

## N999S and D434G cause paroxysmal dyskinesia in mice

One of the most recognizable symptoms in *KCNMA1* channelopathy is a distinctive type of dyskinesia manifesting as sudden, brief paroxysms of axial hypotonia (PNKD3). These episodes sometimes resemble the immobility in non-narcoleptic cataplexy, but have preservation of some muscle tone that varies among individuals (*Du et al., 2005*; *Heim et al., 2020*; *Keros et al., 2022*; *Miller et al., 2021*; *Wang et al., 2017*; *Zhang et al., 2015*). Patients may slump or fall over but can often maintain position if appropriately supported, and consciousness is maintained. Normal activity is recovered

relatively quickly without persistent impairment (see patient videos in *Braverman, 2019*; *Sanders, 2018*). PNKD3 episodes are not initiated by movement or exertion (non-kinesigenic), but rather by negative and positive triggers such as strong emotion (stress and excitement), cold, fatigue, or alcohol. The events are not associated with epileptiform activity on EEG and are generally unresponsive to anti-seizure medications (*Keros et al., 2022*; *Miller et al., 2021*). PNKD3 is associated with substantial morbidity due to its high frequency, with hundreds of episodes per day. All three variants tested in this study are associated with PNKD, with 75% of individuals harboring N999S and D434G carrying the diagnosis but also observed at lower incidence with LOF variants or VUS (variant of uncertain significance) (*Miller et al., 2021*).

There are currently no standardized behavioral assays for either PNKD3 or non-*KCNMA1*-associated PNKD. In other paroxysmal dyskinesia animal models, the phenotype is usually hyperkinetic, not the hypokinetic events observed in PNKD3. For example, $Ca^{2+}$ channelopathy, *Prrt2*-deficient, and *Pnkd* mutant mice are characterized by dystonia, chorea, and tonic-clonic episodes (*Fureman et al., 2002*; *Lee et al., 2012*; *Michetti et al., 2017*; *Tan et al., 2018*; *Pan et al., 2020*). No spontaneous hypotonic dyskinetic motor behavior was detectable to a blinded observer in any of the transgenic lines in this study. Therefore, we sought to elicit episodes by utilizing known triggers for PNKD3. Since individuals harboring N999S and H444Q variants are mostly children without any reported alcohol exposures, a PNKD trigger specifically reported for D434G (*Du et al., 2005*), and no calibratable emotional responses are validated in mice, we focused on the standardizable stress experienced during physical restraint. Stress provocation is the closest stimulus to the natural triggers observed in PNKD3-affected individuals (*Miller et al., 2021*). Restraint stress provoked dyskinesia in most (85%) *tottering* mice (*Fureman et al., 2002*), and *PNKD* mutant mice also showed dyskinesia after stressful handling when placed in a beaker (*Lee et al., 2012*).

To test whether restraint stress would produce paroxysmal dyskinesia, mice were subjected to an acute stereotypical manual restraint protocol by an experienced handler. After restraint, mice were placed into a beaker, a novel-constrained environment proposed to enhance stress (*Lee et al., 2012*). Mice with PNKD-like characteristics were predicted to show restraint-triggered hypokinetic episodes. Mice were scored for abnormal movement (time immobile, circling/hyperactivity, twisting/chorea or limb-clasping, tonic-clonic movement, flattened/dystonic posturing, tremor, listing and falling) in the beaker under video observation. Stereotypical behaviors such as grooming were also recorded. WT mice from all groups showed normal exploratory behavior including sniffing, grooming, and rearing with coordinated movements.

*Kcnma1*[N999S/WT] mice and WT littermates placed in the beaker without prior restraint did not show any dyskinetic movements or collapsing behavior (paws no longer touching the ground). There was no significant difference in the time spent immobile between these groups (*Figure 5A*). Next, restrained mice were placed in the beaker. Since mice increase grooming when released from stress *Shirley et al., 2008*, *Jones and Mormède, 2000*, this behavior was used as a control to indicate the presence of stress. *Kcnma1*[N999S/WT] mice and WT littermates both showed an increase in grooming events after restraint compared to their non-restraint controls (*Figure 5B*), confirming both genotypes responded to stress with an increase in stereotypical behavior.

After stress, *Kcnma1*[WT/WT] mice had exploratory behavior and spent less than a minute immobile in the beaker (51±10 s). Although the range was wider, their time spent immobile did not differ significantly from the unrestrained baseline. In contrast, *Kcnma1*[N999S/WT] mice were immobile for more than twice as long after stress (120±12 s) (*Figure 5A*, *Figure 5—video 1*). After episodes of immobility, both genotypes resumed normal exploratory behavior or grooming.

In qualitative assessment, three *Kcnma1*[N999S/WT] mice had extended myoclonic 'hiccups' throughout the immobility that were not associated with respiratory rate. One mouse also showed listing, and three had a flattened posture. Evaluation of other dyskinetic behaviors (dystonia, chorea, clasping, etc.) in non-restraint controls and after stress revealed grossly normal movements for *Kcnma1*[N999S/WT] mice, with the exception of the notable immobility. In direct comparison, *Kcnma1*[WT/WT] littermates had raised heads and less hunched postures during their briefer immobility, suggesting the maintenance of normal axial tone. Brief hiccups were observed in one WT control, at shorter duration than the *Kcnma1*[N999S/WT] mice, and one mouse had a brief flattened posture during the first bin. Use of a fitted tube restraint, which may produce a stronger stress response, increased the ability of a blinded observer to predict genotype differences in immobility (n=11 mice, data not shown).

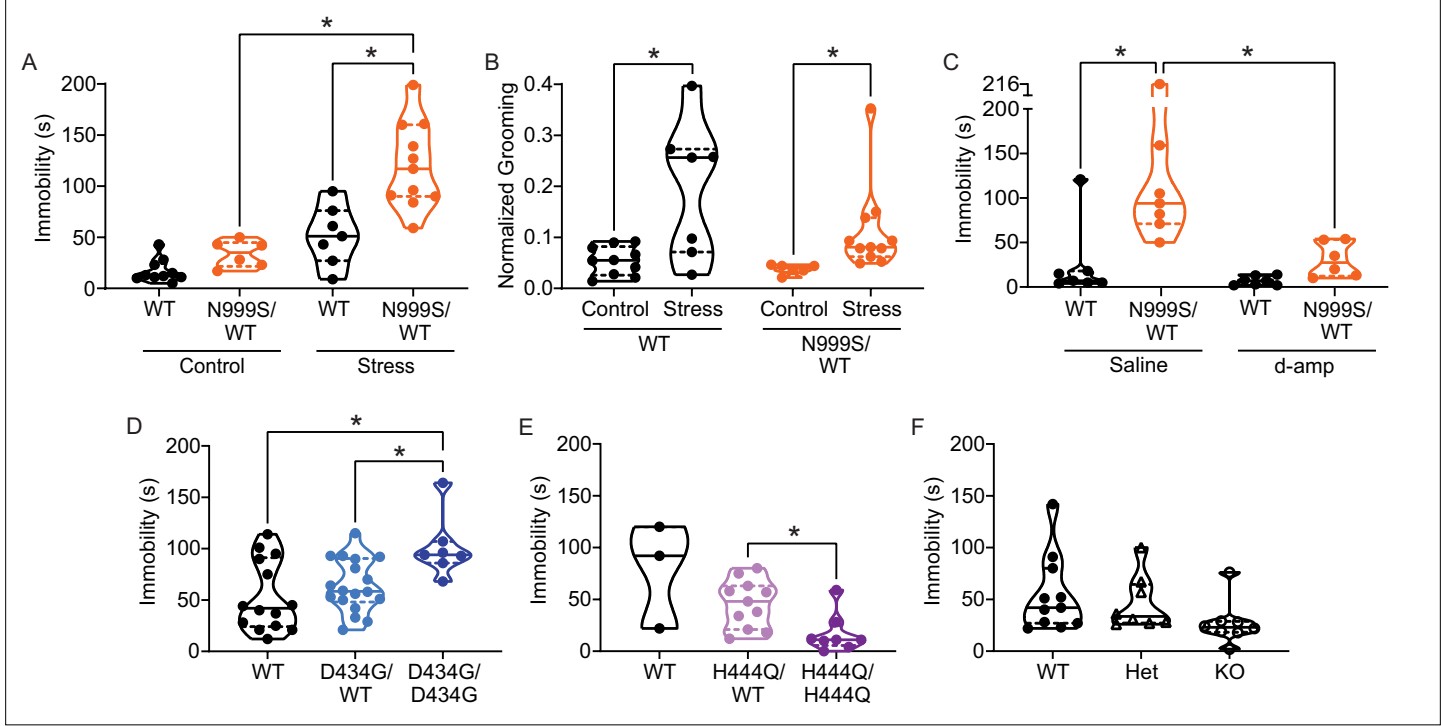

**Figure 5.** Stress-induced paroxysmal dyskinesia. (**A**) Control: Without restraint stress, there was no difference in the time spent immobile between *Kcnma1*[WT/WT] (n=10) and *Kcnma1*[N999S/WT] mice (n=6, p>0.9999; two-way ANOVA with Bonferroni post hoc). Restraint stress: Immobility time was longer for restrained *Kcnma1*[N999S/WT] mice (n=11) compared to *Kcnma1*[WT/WT] (n=7, *p=0.0001; one-way ANOVA), and between restrained *Kcnma1*[N999S/WT] mice (n=11) compared to unrestrained *Kcnma1*[N999S/WT] mice (n=6, *p<0.0001). In contrast, unrestrained *Kcnma1*[WT/WT] mice (n=10) had no differences from restrained *Kcnma1*[WT/WT] mice (n=7, p=0.1174). (**B**) Grooming behavior increased in restrained *Kcnma1*[WT/WT] mice (n=7) compared to unrestrained *Kcnma1*[WT/WT] mice (n=10, *p=0.0300; t-test), and in restrained *Kcnma1*[N999S/WT] mice (n=11) compared to unrestrained *Kcnma1*[N999S/WT] mice (n=6, *p=0.0174; t-test). (**C**) Immobility time was longer for saline-treated *Kcnma1*[N999S/WT] mice (n=7) compared to *Kcnma1*[WT/WT] (n=7, *p=0.0018) and d-amp-treated *Kcnma1*[N999S/WT] mice (n=6, *p=0.0053; two-way ANOVA with Bonferroni post hocs). There was no difference between d-amp-treated *Kcnma1*[WT/WT] mice (n=7), d-amp-treated *Kcnma1*[N999S/WT] mice (n=6, p>0.9999), and saline-treated *Kcnma1*[WT/WT] mice (n=7, p>0.9999). (**D**) After restraint, *Kcnma1*[D434G/D434G] mice (n=7) spent more time immobile compared to *Kcnma1*[WT/WT] mice (n=14, *p=0.0166; one-way ANOVA). However, *Kcnma1*[D434G/WT] mice were not different (n=18, p=0.7174). (**E**) Immobility time was shorter in restrained *Kcnma1*[H444Q/H444Q] mice (n=8) compared to *Kcnma1*[H444Q/WT] mice (n=11, *p=0.0081; t-test). *Kcnma1*[WT/WT] mice were not included in the statistical analysis due to small sample size (n=3). (**F**) *Kcnma1*[−/−] mice (n=8) had reduced immobility compared to *Kcnma1*[−/+] mice (n=8) and *Kcnma1*[+/+] mice (n=11, p=0.0535; Kruskal-Wallis test). Data are individual mice with median and inter-quartile range. Representative videos for this dataset appear in *Figure 5—video 5–1*.

The online version of this article includes the following video, source data, and figure supplement(s) for figure 5:

**Source data 1.** Data file for *Figure 5A–F*.

**Figure supplement 1.** Locomotor, stress, and immobility analysis methodology controls in *Kcnma1*[N999S/WT] mice.

**Figure supplement 1—source data 1.** Data file for *Figure 5—figure supplement 1A–D*.

**Figure 5—video 1.** Restraint stress-induced dyskinesia.

https://elifesciences.org/articles/77953/figures#fig5video1

Six PNKD3-affected individuals harboring the N999S variant have been documented to experience nearly full resolution of immobilizing episodes with lisdexamfetamine, a prodrug of dextroamphetamine (d-amp; *Keros et al., 2022*). D-amp treatment is also highly effective against PNKD3 associated with another GOF *KCNMA1* variant (N536H; *Zhang et al., 2020*). Acute daily d-amp administration reduces the number of PNKD3 episodes during the therapeutic window from >100 to 0 in some cases (*Keros et al., 2022*). If the stress-induced immobile state in *Kcnma1*[N999S/WT] mice is consistent with PNKD3, administering d-amp would be predicted to reduce immobility time.

To test for rescue effects, *Kcnma1*[N999S/WT] mice and WT littermates were injected with low dose d-amp (0.5 mg/kg), similar to PNKD3-affected patients taking lisdexamfetamine or d-amp (*Keros et al., 2022*; *Zhang et al., 2020*). Mice were then subjected to 5 min of restraint stress, followed

with observation of mobility state in the beaker assay (*Figure 5C*). Experimental and separate control assays were conducted during the peak time window reported for d-amp effect from open field locomotion (30 min post-injection; *Fu et al., 2021*; *Gould et al., 2007*). As a control, d-amp administration without restraint stress does not affect immobility in either *Kcnma1*$^{N999S/WT}$ mice or WT littermates (*Figure 5—figure supplement 1A*). This corroborates that after an acute injection, the restraint stress stimulus is still required to produce immobility in *Kcnma1*$^{N999S/WT}$ mice (*Figure 5C*). Furthermore, d-amp does not increase wheel running or increase grooming in either genotype (*Figure 5—figure supplement 1B-C*), similar to prior C57BL/6J studies (*Cytryn, 1980*). These control data further indicate that general locomotor activity and the stress response are not markedly increased under the d-amp dosing conditions.

Under the restraint stress paradigm, saline-injected *Kcnma1*$^{WT/WT}$ mice showed normal exploratory behavior with little immobility (<10 s), with one exception of a single mouse immobilized for 121 s exhibiting a hunched posture. D-amp treatment was not different from saline control for *Kcnma1*$^{WT/WT}$ mice (*Figure 5C*). In contrast, restrained *Kcnma1*$^{N999S/WT}$ mice produced a large increase in immobility after saline injection (>4-fold average), corroborating un-injected animals subjected to restraint stress (*Figure 5A and C*). The majority of saline-injected *Kcnma1*$^{N999S/WT}$ mice spent 1–2 min immobile and assumed a hunched posture with the head lowered.

D-amp-injected *Kcnma1*$^{N999S/WT}$ exhibited little immobility compared to saline-injected *Kcnma1*$^{N999S/WT}$ mice (3-fold less). These mice had normal exploratory behavior with shorter, more frequent episodes of grooming. However, when present, the shorter immobility bouts were associated with hunched posture. After d-amp treatment, there was no statistically significant difference in immobility between *Kcnma1*$^{N999S/WT}$ mice and WT littermates (*Figure 5C*). Automated analysis of the assays by an additional blinded experimenter replicated the finding of abrogation of immobility with d-amp treatment in *Kcnma1*$^{N999S/WT}$ mice (*Figure 5—figure supplement 1D*). In conclusion, these data show that d-amp treatment of *Kcnma1*$^{N999S/WT}$ mice rescues stress-induced immobility at therapeutically relevant doses.

Taken together, these data suggest that the presence of a stressor (restraint) produces a new behavioral state in *Kcnma1*$^{N999S/WT}$ mice (immobility) that was not observed in the absence of the trigger or in WT littermates. If the immobility behavior resulted from stress-induced atonic or absence seizures, these events would likely have been observed during baseline EEG recordings given the number of occurrences in the 5 min beaker assay. However, seizure would not be expected to resolve with low-dose d-amp. Alternatively, if stress-induced immobility resulted from general hypoactivity or altered fear response, open field testing might show a difference in motor exploratory behavior between *Kcnma1*$^{N999S/WT}$ and WT littermates. This was not observed (*Figure 6B*). *Kcnma1*$^{N999S/WT}$ mice were also able to achieve the same peak speed as WT littermates during voluntary wheel running (*Figure 6Aii*). In addition, there is no evidence from patients for correlation of PNKD3 with increased anxiety, depression, or hypoactivity (*Miller et al., 2021*). We conclude that stress-induced immobility, which occurs in brief episodes that are instantaneously recovered, is responsive to d-amp, and occurs without other hyperkinetic or tonic-clonic manifestations, is consistent with the reversible triggered hypokinetic behavioral state in PNKD3-affected individuals (*Heim et al., 2020*; *Keros et al., 2022*).

To further characterize this mouse model for PNKD3, additional motor assays were conducted. Besides stress, PNKD3 episodes can be triggered by positive emotions or excitement, similar to cataplexy in patients with narcolepsy (*Dauvilliers et al., 2014*; *Miller et al., 2021*; *Kelley, 2001Sun et al., 2019a*). Related to the reward and arousal effects in mice, cataplexy can be provoked in narcoleptic orexin-deficient (*Hcrt*$^{-/-}$) mice by wheel running (*España et al., 2007*; *Mahoney et al., 2017*; *Novak et al., 2012*). We also assessed this positive trigger to determine if voluntary wheel running could produce a PNKD-like behavior in the setting of a more complex motor task. In this assay, *Kcnma1*$^{N999S/WT}$ mice covered a shorter distance compared to their WT littermates (*Figure 6Ai*). The reduction in running distance was not due to a decrease in the maximum speed the mice could achieve, but instead occurred as a result of increased gaps in activity (*Figure 6Aii–iv*). These gaps could be consistent with, but not exclusively attributable to, cessation of running during a PNKD-like immobility. However, we cannot rule out the reduced distance as a result of a baseline dyskinesia or muscle weakness, since *Kcnma1*$^{N999S/WT}$ mice also show decreased time to fall in the rotarod and hanging wire assays (*Figure 6C–D*).

PNKD3 is also exhibited in individuals harboring heterozygous D434G variants and is provoked by additional triggers besides stress, such as alcohol (*Du et al., 2005*). However, without diagnostic

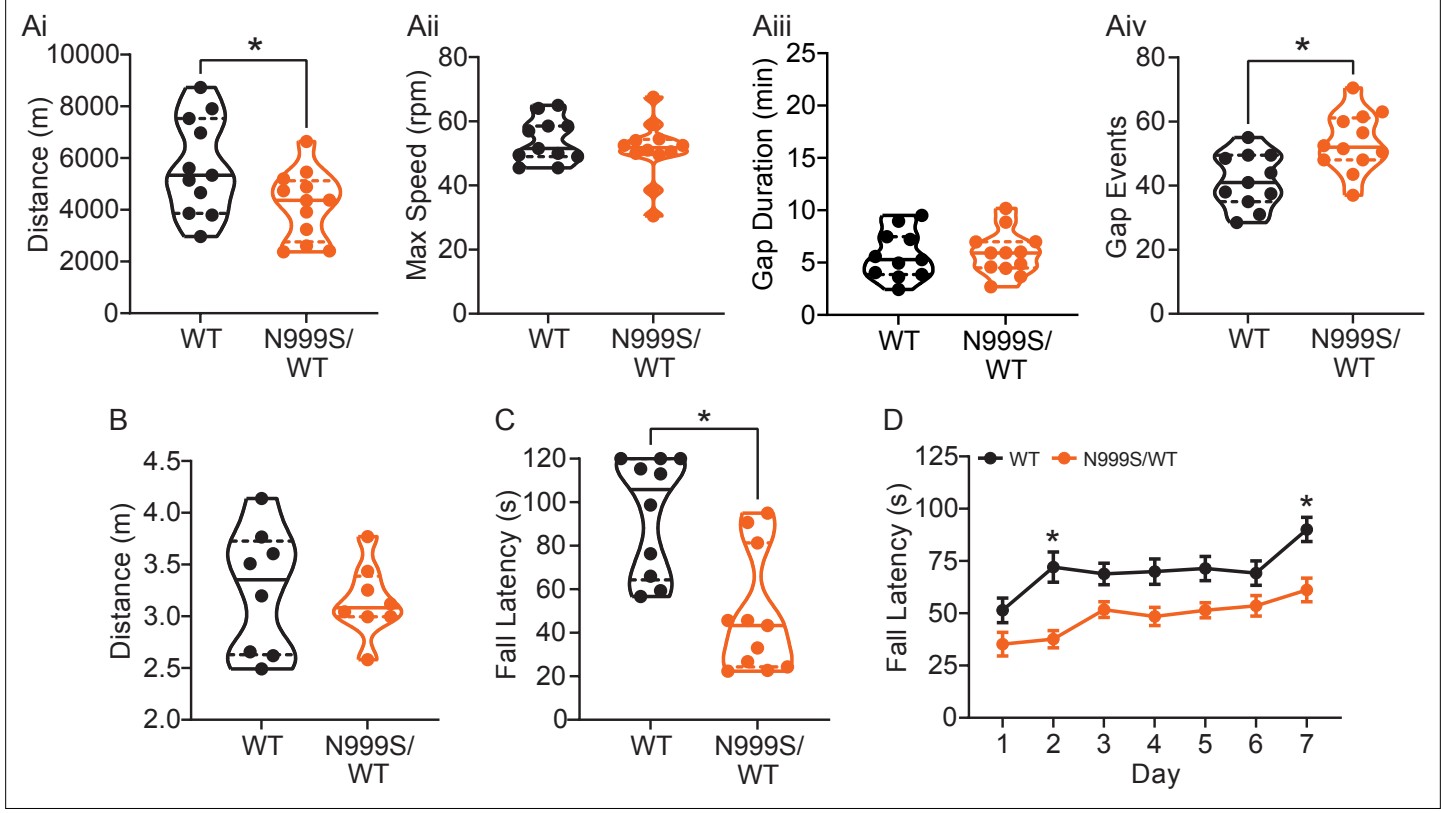

**Figure 6.** Motor coordination in *Kcnma1*[N999S/WT] mice. (**A**) Locomotor wheel running parameters calculated from average activity counts over 48 hr from singly housed mice with free access to wheels. (Ai) Distance covered was reduced for *Kcnma1*[N999S/WT] (n=12) compared to *Kcnma1*[WT/WT] mice (n=11, *p=0.0411; t-test). (Aii) Maximum speed was comparable between *Kcnma1*[N999S/WT] (n=11) and *Kcnma1*[WT/WT] mice (n=12, p=0.3618; t-test). (Aiii) Duration of time off wheels (gap duration) was comparable between *Kcnma1*[N999S/WT] (n=11) and *Kcnma1*[WT/WT] mice (n=12, p=0.8281; t-test). (Aiv) Number of times the mouse was off the wheel (gap events) was higher for *Kcnma1*[N999S/WT] (n=12) compared to *Kcnma1*[WT/WT] mice (n=11, *p=0.0040; t-test). (**B**) Open field assay. *Kcnma1*[N999S/WT] mice (n=8) covered the same distance as *Kcnma1*[WT/WT] mice (n=8) in a 15 min trial (p=0.6973; t-test). (**C**) Acute muscle strength was tested by hanging mice from a stationary platform (cage lid) for 120 s. Fall latency was lower in *Kcnma1*[N999S/WT] (n=11) compared to *Kcnma1*[WT/WT] mice (n=10, *p=0.0014; Mann-Whitney test) indicating weaker grip strength. (**D**) Rotarod assay. Fall latency was lower for *Kcnma1*[N999S/WT] mice (n=11) on day 2 (*p=0.0045) and day 7 (*p=0.0124) compared to *Kcnma1*[N999S/WT] mice (n=12). Motor learning was observable as an improvement in fall latency times across the three trials on each day (data not shown), suggesting the overall impairment was related to motor coordination and not learning. Data are presented as individual data points with median and inter-quartile range (**A–C**) and mean ± SEM (**D**). Results for these assays with *Kcnma1*[D434G], *Kcnma1*[H444Q], and *Kcnma1*[-/-] mice appear in *Figure 6—figure supplements 1 and 2*. For these assays, the baseline motor coordination severity fell in the series *Kcnma1*[-/-]>*Kcnma1*[D434G/D434G] > *Kcnma1*[N999S/WT]>*Kcnma1*[H444Q/H444Q].

The online version of this article includes the following source data and figure supplement(s) for figure 6:

**Source data 1.** Data file for *Figure 6Ai, Aiv, B–DD*.

**Figure supplement 1.** Motor coordination in *Kcnma1*[D434G], *Kcnma1*[H444Q], and *Kcnma1*[-/-] mice.

**Figure supplement 1—source data 1.** Data file for *Figure 6—figure supplement 1Ai, Di, Aii–Dii, Aiii–Diiii*.

**Figure supplement 2.** Hanging wire and rotarod in *Kcnma1*[D434G], *Kcnma1*[H444Q], and *Kcnma1*[-/-] mice.

**Figure supplement 2—source data 1.** Data file for *Figure 6—figure supplement 2Ai–Aiii, Bi–Biiii*.

standardization, it is not clear whether this constitutes a different type of PNKD episode in individuals harboring D434G compared to N999S. In the stress assay, *Kcnma1*[D434G/WT] mice and WT littermates had similar immobility lasting 53±9 and 63±6 s, respectively (*Figure 5D*). However, in homozygous *Kcnma1*[D434G/D434G] mice, immobility time was similar to N999S heterozygotes (101±11 s). Thus, stress-induced dyskinesia is present in the D434G mouse model. It remains possible that alcohol would also be capable of triggering these episodes, but it is difficult to assess given the bi-directional motor effects of alcohol in mice (*Jones and Mormède, 2000*). *Kcnma1*[D434G/D434G] mice have aspects of reduced motor coordination on the rotarod and running wheels but are not hypokinetic under basal conditions (*Figure 6—figure supplements 1 and 2*).

In contrast, homozygous LOF manipulations showed a different directionality in stress-triggered dyskinetic behavior, with less immobility after restraint than either WT or heterozygous littermates. No phenotypic differences were detected in heterozygous mice of either line. However, immobility in *Kcnma1*$^{H444Q/H444Q}$ mice was reduced to 16±6 s (*Figure 5E*), and in *Kcnma1*$^{-/-}$ mice, it was reduced to 27±7 s (*Figure 5F*). Three *Kcnma1*$^{-/-}$ mice also showed hyperactive circling and rapid limb movements notable to the blinded observer, and one had notable tremor during the brief non-active periods. *Kcnma1*$^{H444Q/H444Q}$ and *Kcnma1*$^{-/-}$ mice had reduced latencies to fall from the hanging wire under basal conditions, but only *Kcnma1*$^{-/-}$ mice are profoundly impaired on the rotarod and running wheels due to ataxia (*Figure 6—figure supplements 1 and 2*). The reduced motor performance in *Kcnma1*$^{H444Q/H444Q}$ and *Kcnma1*$^{-/-}$ mice stands in contrast to the reduction in immobility observed after stress (*Figure 5E–F*). These data reveal that the H444Q variant and *KCNMA1* null genotypes are not associated with PNKD immobility under the same triggers that provoke GOF variants. The results raise the possibility that stress-induced dyskinesia manifestation is influenced by mutation type, with GOF producing hypokinetic and LOF producing hyperkinetic responses.

## Discussion

We have characterized the channel properties, neuronal activity, neurobehavioral phenotypes, and relative severity of three *KCNMA1*-linked channelopathy variants under equivalent conditions. Pathogenic potential was established using four criteria (*MacArthur et al., 2014*): (1) low variant frequency in the human population, classifying as a mutation (*Miller et al., 2021*), (2) variant alters BK channel gating properties, (3) variant alters neuronal BK currents and firing, since the channelopathy is a neurological disorder, and (4) variant produces phenotypes similar to the central patient diagnoses—seizure susceptibility and PNKD. The findings support the conclusion that *KCNMA1*-linked channelopathy, although symptomatically heterogenous and comprised predominantly of de novo variants, has the potential to be categorized as a monogenic disorder. The results substantiate hyperexcitability, increased seizure propensity, and PNKD as collective phenotypes replicated in two hypermorphic GOF *KCNMA1* alleles. Moreover, our data for N999S and D434G corroborate mounting evidence, both in patients (*Du et al., 2005*; *Keros et al., 2022*; *Miller et al., 2021*; *Zhang et al., 2020*) and in animal models (*Dong et al., 2021*; *Kratschmer et al., 2021*), that PNKD can be considered the most consistent symptom for *KCNMA1* GOF channelopathy.

N999S produced the strongest GOF effect on BK channel properties under these study conditions. Although N999S is the most commonly reported *KCNMA1* variant, direct evidence that it caused channelopathy was lacking because it arose de novo in all known cases (*Keros et al., 2022*; *Miller et al., 2021*). In mice, our data validate its dominant inheritance and pathogenic potential as a GOF mutation able to increase BK current and neuronal activity in the heterozygous configuration in mice. Neurobehavioral validation further identified increased PTZ-induced seizure propensity and stress-triggered dyskinesia episodes resembling PNKD-like immobility, broadly consistent with the phenotypic occurrence in patients. The lethality of homozygous *Kcnma1*$^{N999S/N999S}$ (and hemizygous *Kcnma1*$^{N999S/\Delta}$) genotypes, which have not been found in any patient, underscore the severity of this variant. It is not known which cell type the lethality stems from, but N999S channels can produce both increased peak action potential-evoked and subthreshold BK current (*Moldenhauer et al., 2020b*). Thus it is possible that N999S could increase firing in some cells, exemplified by the effect in dentate granule neurons, or prevent firing, depending on the cellular context.

D434G, a less severe GOF mutation than N999S at the BK channel level, had dominant inheritance for a subset of traits in our model, partially paralleling the familial pedigree (*Du et al., 2005*). The increased BK current, excitability, and PTZ-induced seizure propensity in the heterozygous configuration validated D434G pathogenicity. However, homozygosity was required to produce the PNKD-like immobility attacks, ranking D434G less pathogenic than N999S. While *Kcnma1* $^{D434G/D434G}$ mice would produce obligate homotetrameric mutant BK channels, additional work is needed to understand whether the decreased severity in *Kcnma1*$^{D434G/WT}$ mice is indicative of heterotetrameric (WT:D434G) BK channel formation. Interestingly, three other D434G models have been reported, with some phenotypic variation. BAC-loxP-D434G mice exhibited GTC seizures in the absence of motor dysfunction (*Ling et al., 2016*). Another Cre/lox-D434G mouse line was comparatively more severe with complete penetrance of absence seizures and dyskinesia in the heterozygous configuration (*Dong et al., 2021*). This variability, observed in both mouse models and incomplete penetrance

in patients, raises the possibility that additional genetic or environmental factors can influence symptomatic severity. Nevertheless, an analogous mutation in flies also alters neuronal activity and baseline motor behavior (*Kratschmer et al., 2021*).

The LOF variant, H444Q, demonstrated limited pathogenicity by decreasing BK channel activity but was not validated as a hypomorphic or haploinsufficient allele in neurons or mice. Homozygous *Kcnma1*[H444Q/H444Q] mice showed a neurobehavioral phenotype distinct from PNKD: stress-induced hyperkinetic motor responses similar to *Kcnma1*[−/−] null mice. Given multiple genetic findings and symptomatic ambiguity in the patient carrying this variant, the different dyskinetic responses compared to the two GOF models may suggest a basis to improve the diagnostic investigations for this and other *KCNMA1* variants classified as LOF or VUS. At present, there are no patients with homozygous *KCNMA1* alleles validated as functionally null for channel activity (*Miller et al., 2021*), but the ataxia, tremor, decreased strength, and hyperactivity in *Kcnma1*[−/−] mice (*Imlach et al., 2008*; *Meredith et al., 2004*; *Meredith et al., 2006*; *Sausbier et al., 2004*; *Typlt et al., 2013*; *Wang et al., 2020*) are symptoms observed at lower incidence among patients. Lastly, our experimental conditions failed to corroborate the influence of LOF alleles on seizure propensity predicted from several animal studies (*Ermolinsky et al., 2008*; *Kuebler et al., 2001*; *Pacheco Otalora et al., 2008*; *Sheehan et al., 2009*; *Shruti et al., 2008*; *Yao et al., 2021*). *KCNMA1* LOF channelopathy has also been proposed to carry a broader set of non-overlapping features associated with a subset of de novo LOF variants, referred to as Liang-Wang syndrome (*Liang et al., 2019*). However, none of the observable patient correlates were present in the LOF H444Q model studied here. H444Q mice also had little overlap with *Ryegrass Staggers*, a toxicity syndrome of livestock involving BK channel inhibition that is phenotypically similar to *Kcnma1*[−/−] mice (*Imlach et al., 2008*).

Genotype-phenotype relationships are important for understanding *KCNMA1* channelopathy disease mechanisms as well as potential therapeutics. The allelic series established from N999S, D434G, and H444Q BK channels in vitro is outwardly congruent with the relative severity in mice. Such allelic series have been pivotal in understanding other complex channelopathies, especially those delineating distinct disorders within the same gene association (*Noebels, 2003*; *Pietrobon, 2005*; *Zwingman et al., 2001*). However, it is not yet clear whether all *KCNMA1* variants discovered in the setting of neurological diagnoses carry the same pathogenic potential. Because this *KCNMA1* allelic series was derived from a limited set of conditions, designations of phenotypic severity could be further influenced by additional factors.

At the DNA level, it is likely that all three variant-containing alleles are expressed because the homozygous phenotypes are dissimilar to *Kcnma1*[−/−]: *Kcnma1*[N999S/N999S] and *Kcnma1*[D434G/D434G] have unique phenotypes not found in *Kcnma1*[−/−] (lethality and stress-induced immobility, respectively), and *Kcnma1*[H444Q/H444Q] had fewer and less severe phenotypes than *Kcnma1*[−/−] (e.g., rotarod). However, the phenotypic severity for the human variants could be mitigated by their context within the mouse gene, which was not humanized through additional rounds of gene editing. Mouse and human BK channels differ at eight constitutive coding residues and have minor differences in BK current properties in heterologous cells (*Lai, 2015*). D434G produces a larger GOF effect on the G-V relationship in the context of a human BK channel compared to mouse (*Wang et al., 2009*). Similarly, as the least potent variant, it is possible that H444Q has a further reduced effect on BK channel properties in mouse, potentially contributing to the lack of BK current differences.

At the channel level, unresolved questions concerning functional mechanisms and subunit composition illustrate the influence of additional factors besides variant genotype on the phenotypic severity. For example, BK current levels were relatively similar between *Kcnma1*[N999S/WT] and *Kcnma1*[D434G/WT] in one type of neuron (dentate granule), yet these two heterozygous genotypes exhibited differences in excitability. Instead, homozygous *Kcnma1*[D434G/D434G] firing curves were more comparable to *Kcnma1*[N999S/WT]. Underlying this, the ratio of expression from WT and mutant alleles, stoichiometry and properties of heterotetrameric channels, alternative splice variation, and the composition of α:β:$Ca_V$ macrocomplexes in the loci responsible for neuropathology in heterozygous transgenic mice all still remain to be resolved. Additional experiments to probe the voltage and $Ca^{2+}$-dependent bases for N999S and D434G gating defects (*Diez-Sampedro et al., 2006*; *Du et al., 2005*; *Li et al., 2018*; *Moldenhauer et al., 2020b*; *Yang et al., 2010*), and the manifestation in the presence of the highly expressed β4 subunit (*Wang et al., 2009*; *Berkefeld and Fakler, 2013*; *Wang et al., 2016*), may also further explain these relative differences in BK currents and pathogenicity. Interestingly, no *KCNMB4*

variants associated with seizure have been described yet, even as the number of seizure-associated BKα variants has increased (*Miller et al., 2021*). This could suggest that loss of β4 regulation would have more severe consequences.

At the neurobehavioral level, increased dentate granule cell excitability may contribute directly or indirectly to the reduced seizure thresholds, but other areas of the brain are additionally involved in the generalized PTZ-evoked seizures. The BK channel inhibitor paxilline can block chemoconvulsant-induced seizures, associated with changes in cortical excitability (*Sheehan et al., 2009*; *Shruti et al., 2008*). These data suggest additional neurons to investigate for links between GOF BK channels and hyperexcitability. It is also not clear yet how PNKD-like symptomology is produced. BK channels regulate excitability in several motor-associated regions including the cerebellum, striatum, neuro-muscular junction, and skeletal muscle (*Abrahao et al., 2017*; *Goldberg and Wilson, 2005*; *Sausbier et al., 2005*; *Tricarico et al., 1997*; *Vatanpour and Harvey, 1995*; *Wang et al., 2020*). Cre/lox-D434G and *PRRT2* mice show changes in cerebellar excitability and morphology (*Calame et al., 2020*; *Dong et al., 2021*) but do not show hypokinetic PNKD described in this study. In other PNKD animal models, as in patients, the brain mechanisms are not well understood. Some PNKDs respond to anti-epileptic medications, a few respond to deep-brain stimulation, but most necessitate trigger avoid-ance (*Manso-Calderón, 2019*). In PNKD3, stimulants (lisdexamfetamine and dextroamphetamine) are highly effective in reducing attacks described as both dystonic and hypotonic (*Keros et al., 2022*; *Zhang et al., 2020*), but neither drug has a direct effect on BK$^{WT}$ channel activity (*Figure 1—figure supplement 2* and *Zhang et al., 2020*) or BK$^{N999S}$ (*Figure 1—figure supplement 2*) in heterologous cells, leaving the target for their actions on PNKD3 an open question. The significant reduction in stress-induced immobility with d-amp treatment in *Kcnma1*$^{N999S/WT}$ mice substantiates the utility of this model in future studies dissecting the cell and circuit basis for the PNKD3. Clinical observations and d-amp responsiveness in patients predict a central neurological dysfunction in producing the debilitating hypokinetic state (*Keros et al., 2022*; *Zhang et al., 2020*), but whether other factors such as altered neuromuscular transmission or episodic muscle hypotonia make some contribution during attacks remains to be systematically tested.

Changes in BK channel function and/or *KCNMA1* expression are associated with a growing number of neurodevelopmental disorders including epilepsy, dyskinesia, autism, Angelman's syndrome, Fragile X syndrome, and brain and skeletal malformations (*Cheng et al., 2021*; *Deng and Klyachko, 2016*; *Du et al., 2020*; *Kessi et al., 2020*; *Kshatri et al., 2020*; *Laumonnier et al., 2006*; *Liang et al., 2019*; *Miller et al., 2021*; *N'Gouemo, 2014*; *Sun et al., 2019a*; *Sun et al., 2019b*). Neuropathology in these disorders is associated with changes in BK channel activity in both directions. Yet it has been challenging to distill *KCNMA1*-linked channelopathy into a cohesive GOF versus LOF symptomology because the existing patient data lack genetic pedigrees and diagnostic cross-comparability. Looking ahead, the phenotypic penetrance and heterogeneity investigated here validate only a few of the 40+ patient-associated *KCNMA1* variants, but it will not be possible to make transgenic models for every case. There is less symptomatic consistency among non-GOF alleles (LOF or VUS), identifying this as a potentially fruitful area for future investigations.

## Materials and methods

### Key resources table

| Reagent type (species) or resource | Designation | Source or reference | Identifiers | Additional information |
|---|---|---|---|---|
| Recombinant DNA reagent | BK$^{N999S}$ | This paper | MG279689 with rs886039469 | BK channel expression construct in pcDNA3.1+ (*Figure 1*) |
| Recombinant DNA reagent | BK$^{D434G}$ | This paper | MG279689 with rs137853333 | BK channel expression construct in pcDNA3.1+ (*Figure 1*) |
| Recombinant DNA reagent | BK$^{H444Q}$ | This paper | MG279689 with c.1332C>G, p.H444Q | BK channel expression construct in pcDNA3.1+ (*Figure 1*) |
| Recombinant DNA reagent | BK$^{WT}$ | Genbank | hBK$_{QEERL}$ MG279689 | BK channel expression construct in pcDNA3.1+ (*Figure 1*) |
| Cell line (*Homo sapiens*) | HEK293T | ATCC | CRL-11268 | |

*Continued on next page*

*Continued*

| Reagent type (species) or resource | Designation | Source or reference | Identifiers | Additional information |
|---|---|---|---|---|
| Chemical compound, drug | DMEM | Gibco, Life Technologies Corp | Cat. #11995-065 | |
| Chemical compound, drug | Fetal bovine serum | Sigma-Aldrich | Cat. #4135 | |
| Chemical compound, drug | Penicillin/streptomycin | Mediatech Inc | Cat. #30-002 Cl | |
| Chemical compound, drug | L-glutamine | Mediatech Inc | Cat. #25-005 Cl | |
| Chemical compound, drug | Trans-IT LT1 | Mirius Biological | | |
| Chemical compound, drug | Poly-L-lysine | Sigma-Aldrich | Cat. #P4832 | |
| Chemical compound, drug | Dextroamphetamine sulfate | Sigma-Aldrich | Cat. #1180004 | |
| Gene (*Mus musculus*) | Kcnma1 | Gene Bank; Ensembl | ID: 16531; ENSMUSG00000063142 | |
| Strain (*Mus musculus*) | C57BL/6J | Jackson Laboratories | Stock #000664 | |
| Genetic reagent (*Mus musculus*) | *Kcnma1*$^{N999S}$ | This paper | Gene ID:16531 with rs886039469 | Mouse line maintained in A. Meredith's lab (*Figure 1—figure supplement 1*) |
| Genetic reagent (*Mus musculus*) | *Kcnma1*$^{D434G}$ | This paper | Gene ID:16531 with rs137853333 | Mouse line maintained in A. Meredith's lab (*Figure 1—figure supplement 1*) |
| Genetic reagent (*Mus musculus*) | *Kcnma1*$^{H444Q}$ | This paper | Gene ID:16531 with c.1332C>G, p.H444Q | Mouse line maintained in A. Meredith's lab (*Figure 1—figure supplement 1*) |
| Sequence-based reagent (oligonucleotides) | N999S gRNA | Integrated DNA Technologies | CTGTATGAAGT TACTGTTAT | |
| Sequence-based reagent (oligonucleotides) | D434G/H444Q gRNA | Integrated DNA Technologies | GGACCGGGATGA TGTCAACG | |
| Sequence-based reagent (oligonucleotides) | N999S donor | Integrated DNA Technologies | AGATACTAAGAAAA GTTGTAATTTGGAC ATCAATTGTGATTTT CGGTGTTGGCTTAA GAATGCTTCTCTTC TACCTTCTTT CTCC AGACATAtTTC AgTGACAATATtCTCA CCCTAATACGGACCC TGGTGACAGGAGGAG CCACACCA | |
| Sequence-based reagent (oligonucleotides) | D434G donor | Integrated DNA Technologies | CTCTGGAGAGTGTCT CTAACTTCCTGAAGG ACTTTCTGCACAAGG ACCGtGgTGATGTCAA CGTtGAGATTGTCTTT CTTCACAAGTAAGAGC CCCCTGCTGCCACCA GACCCTGCCACC | |
| Sequence-based reagent (oligonucleotides) | H444Q donor | Integrated DNA Technologies | CTCAGAGAGAAGCAT GAGTTTAGGTGGCAG GGTCTGGTGGCAGCA GGGGGCTCTTACTTcT GcAGAAAGACgAT CTCgACGTTGACATC ATCCCGGTCCTTGTG CAGAAAGTCCTTCAGG | |
| Sequence-based reagent (oligonucleotides) | N999S genotyping primer (F) | Transnetyx, Inc | TCGGTGTTGGCTTA AGAATGCTT | *Kcnma1*$^{N999S}$ |
| Sequence-based reagent (oligonucleotides) | N999S genotyping primer (R) | Transnetyx, Inc | CCTCAGCTATTAGAG CCTCGAGCTC | *Kcnma1*$^{N999S}$ |
| Sequence-based reagent (oligonucleotides) | WT genotyping reporter | Transnetyx, Inc | CAGACATACTTCAAT GACAATAT | *Kcnma1*$^{N999S}$ |
| Sequence-based reagent (oligonucleotides) | N999S genotyping reporter | Transnetyx, Inc | CAGACATATTTCAGT GACAATAT | *Kcnma1*$^{N999S}$ |

*Continued on next page*

*Continued*

| Reagent type (species) or resource | Designation | Source or reference | Identifiers | Additional information |
|---|---|---|---|---|
| Sequence-based reagent (oligonucleotides) | D434G genotyping primer (F) | Transnetyx, Inc | CTCTAACTTCCTGAA GGACTTTCTGCACA | *Kcnma1*[D434G] |
| Sequence-based reagent (oligonucleotides) | D434G genotyping primer (R) | Transnetyx, Inc | CAGAGAGAAGCATG AGTTTAGGTGGCA | *Kcnma1*[D434G] |
| Sequence-based reagent (oligonucleotides) | WT genotyping reporter | Transnetyx, Inc | ACCGGGATGATGTCA | *Kcnma1*[D434G] |
| Sequence-based reagent (oligonucleotides) | D434G genotyping reporter | Transnetyx, Inc | ACCGTGGTGATGTCAA | *Kcnma1*[D434G] |
| Sequence-based reagent (oligonucleotides) | H444Q genotyping primer (F) | Transnetyx, Inc | CTGTGGACACATTAC TCTGGAGAGTG | *Kcnma1*[H444Q] |
| Sequence-based reagent (oligonucleotides) | H444Q genotyping primer (R) | Transnetyx, Inc | GGGTCTGGTGGCAGCA | *Kcnma1*[H444Q] |
| Sequence-based reagent (oligonucleotides) | WT genotyping reporter | Transnetyx, Inc | TCTTACTTGTGAAGAAAG | *Kcnma1*[H444Q] |
| Sequence-based reagent (oligonucleotides) | H444Q genotyping reporter | Transnetyx, Inc | CTCTTACTTCTGCAGAAAG | *Kcnma1*[H444Q] |
| Genetic reagent (*Mus musculus*) | *Kcnma1*[−/+] | PMID:15184377 DOI: 10.1074/jbc.M405621200 or available from the Jackson Laboratories | *Slo1*[−/−] (Meredith Lab) or Stock #035902 (B6.129(FVB)-*Kcnma1*[tm1Rwa]/J, Jackson Laboratories) | Breeder to generate *Kcnma1*[−/−] |
| Sequence-based reagent (oligonucleotides) | WT genotyping primer (F) | Transnetyx, Inc | CATCATACCGGTGACCATGGA | *Kcnma1*[−/−] |
| Sequence-based reagent (oligonucleotides) | WT genotyping primer (R) | Transnetyx, Inc | CCAAGAAAGCCCACCACATG | *Kcnma1*[−/−] |
| Sequence-based reagent (oligonucleotides) | WT genotyping reporter | Transnetyx, Inc | CCCGGCTGTCGCACG | *Kcnma1*[−/−] |
| Sequence-based reagent (oligonucleotides) | Neomycin genotyping primer (F) | Transnetyx, Inc | GGGCGCCCGGTTCTT | *Kcnma1*[−/−] |
| Sequence-based reagent (oligonucleotides) | Neomycin genotyping reporter | Transnetyx, Inc | CCTCGTCCTGCAGTTCATTCA | *Kcnma1*[−/−] |
| Sequence-based reagent (oligonucleotides) | Neomycin genotyping primer (R) | Transnetyx, Inc | ACCTGTCCGGTGCCC | *Kcnma1*[−/−] |
| Commercial assay, kit | miRNeasy Mini Kit | Qiagen | Cat. #217004 | |
| Commercial assay, kit | Mouse Clariom D Assay | Applied Biosystems | Cat. #902514 | |
| Chemical compound, drug | Paxilline | alomone labs | Cat. #P-450 | |
| Chemical compound, drug | 4-Aminopyridine | Sigma-Aldrich | Cat. #275875 | |
| Chemical compound, drug | TTX | alomone labs | Cat. #T-550 | |
| Chemical compound, drug | Pentylenetetrazol | Sigma-Aldrich | Cat. #P6500 | |
| Software, algorithm | pClamp10.7 | Molecular Devices | | |
| Software, algorithm | Transcriptome Analysis Console Software | ThermoFisher Scientific | TAC version 4.0.1 | |
| Software, algorithm | Sirenia Acquisition software | Pinnacle Technology Inc | Version 2.2.4 | |
| Software, algorithm | Sirenia Seizure Pro software | Pinnacle Technology Inc | Cat. #9037 | |
| Software, algorithm | Prism Software | GraphPad (Dotmatics) | Prism version 9.02 | |
| Software, algorithm | Ethovision Software | Noldus Information Technology | Ethovision XT version 11.5 | |

## HEK cell patch-clamp electrophysiology

The N999S (rs886039469; also numbered as N995S, N1036S, and N1053S in other reference sequences), D434G(rs137853333), and H444Q mutations were introduced into wild-type (WT) human

hBK$_{QEERL}$ cDNA sequence (MG279689) in the pcDNA3.1+ mammalian expression vector. Mutations were verified by sequencing. Channel constructs contained an N-terminal Myc tag and an EYFP tag in regulators of conductance of potassium 2 (RCK2) domain.

HEK293T cells (CRL-11268, ATCC, Manassas, VA) were cultured in media containing Dulbecco's modified Eagle medium (Cat. #11995-065, Gibco, Life Technologies Corp., Grand Island, NY), 10% fetal bovine serum (Cat. #4135, Sigma-Aldrich, St. Louis, MO), 1% penicillin/streptomycin (Cat. #30-002 Cl, Mediatech Inc, Manassas, VA), and 1% L-glutamine (Cat. #25-005 Cl, Mediatech Inc, Manassas,VA) and incubated with 5% carbon dioxide at 37°C. HEK cells were transfected with WT or mutant constructs using Trans-IT LT1 (Mirius Biological, Madison, WI) at 1:2 ratio of DNA and the reagent. After 4–12 hr, cells were re-plated onto glass coverslips pre-coated with poly-L-lysine (Cat. #P4832, Sigma-Aldrich, St Louis, MO). After 14–24 hr, recordings were performed. Constructs were allocated into the same cell transfections on the same days without blinding. HEK cells were authenticated at 17 STR loci with 94% profile match; Mycoplasma negative (ATCC Authentication Service).

BK currents were recorded using inside-out patch-clamp at room temperature in symmetrical K$^+$. One μM intracellular Ca$^{2+}$ was used, a physiologically relevant Ca$^{2+}$ condition near concentrations where altered gating behavior was manifested in prior studies (*Li et al., 2018*; *Moldenhauer et al., 2020b*; *Moldenhauer et al., 2020a*; *Wang et al., 2009*; *Yang et al., 2010*). Thin-walled borosilicate glass pipettes with resistances of 1–3 MΩ were filled with (in mM): 140 KMeSO$_3$, 2 KCl, 2 MgCl$_2$, and 20 HEPES. The internal (bath) solution contained (in mM): 140 KMeSO$_3$, 2 KCl, 20 HEPES, and 5 HEDTA with CaCl$_2$ added to achieve 1 μM free CaCl$_2$, pH adjusted to 7.2 with KOH. Free Ca$^{2+}$ concentrations were calculated with WebMaxC: (https://somapp.ucdmc.ucdavis.edu/pharmacology/bers/maxchelator/webmaxc/webmaxcS.htm).

Macroscopic currents were recorded with a Multiclamp 700B amplifier, and signals were filtered at 10 kHz and digitized at 50 kHz using Digidata1440A and pCLAMP v10 software (Molecular Devices, Sunnyvale, CA). BK currents were activated with a voltage protocol stepped from a holding potential of −100 mV stepping to +250 mV with +10 mV increments for 30 ms and back to −100 mV for 15 ms to generate tail currents. G-V curves were generated from the tail currents 150–200 μs after the peak normalized to the maximum conductance (G/G$_{max}$) and plotted against the activating voltage step (V). V$_{1/2}$ values were calculated from a Boltzmann fit of the G-V curves (Prism v9 GraphPad Software, San Diego, CA). Leak currents were compensated using a P/5 protocol with a subsweep holding potential of −120 mV.

Activation time constants were obtained from the same patches in *Figure 1*. $\tau_{act}$ was obtained by fitting the rising phase of the outward K$^+$ current to single exponential function. For the deactivation kinetics, BK currents were elicited by +200 mV voltage command for 20 ms from a holding potential of −100 mV followed by 15 ms voltage steps from −200 to −10 mV with +10 mV increments. Deactivation time constants were obtained by fitting tail currents with single exponential functions. Leak currents were compensated using a P/5 protocol with a subsweep holding potential of −120 mV.

For experiments with dextroamphetamine and lisdexamfetamine, BK currents were recorded in inside-out patches in physiological K$^+$ and 10 μM intracellular Ca$^{2+}$ as described in *Moldenhauer et al., 2020a*. In voltage-clamp mode, patches were held at −150 mV, stepped from −150 to +150 mV for 30 ms (10 mV increments), and stepped back to −150 mV. Lisdexamfetamine dimesylate (catalog L-026, Supelco Millipore-Sigma) and dextroamphetamine (catalog 1180004, Millipore Sigma) were applied at 155 ng/ml, and paxilline (#2006; Tocris, Bristol, UK) was applied at 100 ng/ml. Current levels were assessed at baseline and 5 min after drug application and were normalized to control current levels for each patch.

### Generation of *Kcnma1*$^{N999S}$, *Kcnma1*$^{D434G}$, and *Kcnma1*$^{H444Q}$ mouse lines

Heterozygous founders introducing N999S (AAT→ A**G**T, exon 25), D434G (GAT→ G**G**T, exon 10), and H444Q (CAC → CA**G**, exon 10) mutations into the mouse *Kcnma1* gene (Gene ID:16531) were generated with CRISPR/Cas9 homologous recombination methods in the C57BL/6J strain (*Figure 1—figure supplement 1*). *Kcnma1*$^{D434G/WT}$ and *Kcnma1*$^{H444Q/WT}$ mice were generated at the Transgenic Mouse Core at John Hopkins University (Baltimore, MD). *Kcnma1*$^{N999S/WT}$ mice were generated at the Jackson Laboratory (Bar Harbor, ME). Transgenic mice were validated with *Kcnma1* sequencing, and founders without additional non-synonymous mutations were bred with C57BL/6J for N1 progeny at The University of Maryland School of Medicine. Genotyping was performed at Transnetyx, Inc

(Cordova, TN) using primers described in 'Genotyping of Kcnma1$^{N999S}$, Kcnma1$^{D434G}$, Kcnma1$^{H444Q}$, and Kcnma1$^{-/-}$ mouse lines' section. Transgenic lines were backcrossed with C57BL/6J up to four generations (N4) for experimental cohorts with heterozygous progeny.

N2-N4 heterozygous mice were intercrossed to produce homozygous progeny. Transgenic heterozygous Kcnma1$^{N999S/WT}$, Kcnma1$^{D434G/WT}$, Kcnma1$^{H444Q/WT}$ (as well as homozygous Kcnma1$^{D434G/D434G}$ and Kcnma1$^{H444Q/H444Q}$) mice showed no gross differences in home cage behavior, body weights or gross morphology, or breeding. Kcnma1$^{N999S/WT}$ × Kcnma1$^{N999S/WT}$ heterozygous crosses produced either no viable pups (n=5/10 breedings) or no homozygous pups (n=56 pups). Additional crosses with Kcnma1$^{-/+}$ heterozygous dams mated to Kcnma1$^{N999S/WT}$ males (n=15 viable pups from five breedings) also produced no Kcnma1$^{N999S/-}$ pups.

Kcnma1$^{-/-}$ (Slo KO) mice were maintained on a C57BL/6J background (>N12 generation). Littermates of each genotype were produced from heterozygous Kcnma1$^{+/-}$ × Kcnma1$^{+/-}$ breeding pairs, as described previously (Meredith et al., 2004) using primer sequences described in 'Genotyping of Kcnma1$^{N999S}$, Kcnma1$^{D434G}$, Kcnma1$^{H444Q}$, and Kcnma1$^{-/-}$ mouse lines' section. For all lines, male and female mice were separated by sex and group housed on a 12 hr light/12 hr dark cycle. Chow and water were provided ad libitum. Kcnma1$^{N999S/WT}$ breeders were fed with high fat chow supplement to enhance breeding. All experiments were conducted in accordance with the University of Maryland School of Medicine Animal Care and Use Guidelines and approved by the Institutional Animal Care and Use Committee (Protocol #1120011). Sex-matched WT and transgenic littermates of both sexes were used for experimental procedures at the indicated ages. Experimenters were blinded to animal genotype at data collection and analysis.

## Genotyping of Kcnma1$^{N999S}$, Kcnma1$^{D434G}$, Kcnma1$^{H444Q}$, and Kcnma1$^{-/-}$ mouse lines

Genotyping was performed tail snips by TaqMan real-time PCR at Transnetyx, Inc (Cordova, TN) using the following: Kcnma1$^{N999S}$ (Kcnma1-9 MUT probe set: (F) TCGGTGTTGGCTTAAGAATGCTT; (R) CCTCAGCTATTAGAGCCTCGAGCTC; WT reporter: CAGACATACTTCAATGACAATAT; N999S reporter: CAGACATATTTCAGTGACAATAT), Kcnma1$^{D434G}$ (Kcnma1-8 MUT probe set: (F) CTCTAACT TCCTGAAGGACTTTCTGCACA; (R) CAGAGAGAAGCATGAGTTTAGGTGGCA; WT reporter: ACCG GGATGATGTCA; D434G reporter: ACCGTGGTGATGTCAA), and Kcnma1$^{H444Q}$ (Kcnma1-7 MUT probe set: (F) CTGTGGACACATTACTCTGGAGAGTG; (R) GGGTCTGGTGGCAGCA; WT reporter: TCTT ACTTGTGAAGAAAG; H444Q reporter: CTCTTACTTCTGCAGAAAG).

Kcnma1$^{-/-}$ (Slo KO) mice were genotyped using the WT primer set (F) CATCATACCGGTGACC ATGGA; (R) CCAAGAAAGCCCACCACATG; WT Reporter: CCCGGCTGTCGCACG and Neomycin primer set (F) GGGCGCCCGGTTCTT; (R) CCTCGTCCTGCAGTTCATTCA; Neo Reporter: ACCTGTCC GGTGCCC.

## Patient data

Patient phenotype and genetics data cited as 'data not shown' was exempt under University of Maryland School of Medicine Institutional Review Board (IRB) Non-Human Subjects Research (NHSR) Protocols HP-00083221, HP-00086440, and HP-00092434.

## Gene expression

Unilateral hippocampus and the medial portion of cerebellum were extracted from 4-month-old mice, and directly put them in 1.0 mm diameter zirconium beads with 750 µl of trizol for bead homogenization. RNA was extracted using the miRNeasy Mini Kit (Qiagen, Valencia, CA) following the manufacturer's protocol. To assess the RNA quality, RNA was quantified via RNA-40 nanodrop and OD 260/280 ratio of all samples were between 1.94 and 2.05. Integrity of RNA was examined via Agilent 2100 Bioanalyzer (Agilent Technologies, Palo Alto, CA). RNA integrity number of all samples were >9. For transcriptome analysis, mouse Clariom D Assay (Applied Biosystems, Waltham, MA) was used following the manufacturer's instruction. RNA extraction and array processing were done at the Genomics Core Facility, University of Maryland, Baltimore, MD. The raw microarray profiling data was preprocessed and quartile-normalized using the Transcriptome Analysis Console Software (version 4.0.1) (accessed on December 1, 2021). All samples passed

array quality control evaluation. Data normalization, differential expression, and hierarchical clustering analysis was performed with default parameters by an experimenter blinded to genotype and brain region.

## Hippocampal slice electrophysiology

Three- to 4-week-old mice were anesthetized with isoflurane, and brains were removed and placed into ice-cold sucrose dissection solution (in mM): 10 $MgCl_2$, 26 $NaHCO_3$, 1.25 $Na_2HPO_4$, 3.5 KCl, 0.05 $CaCl_2$, 10 glucose, 200 sucrose, 1.2 sodium pyruvate, and 0.4 vitamin C, bubbled with 95% $O_2$ and 5% $CO_2$. The brain slices were cut coronally at 300 μm on a VT1000S vibratome (Leica Microsystems, Wetzlar, Germany) at 3–4°C. Slices containing hippocampus were incubated at 32°C for 30 min and kept at the room temperature in oxygenated artificial cerebrospinal fluid (ACSF) containing (in mM): 125 NaCl, 1.7 $MgCl_2$, 26 $NaHCO_3$, 1.25 $Na_2HPO_4$, 3.5 KCl, 2 $CaCl_2$, 10 glucose, 1.2 sodium pyruvate, and 0.4 vitamin C.

Whole-cell patch clamp recordings were performed in the dentate gyrus granule cells of the hippocampus using borosilicate glass electrodes pulled at 3–5 MΩ filled with (in mM): 123 K-MeSO₃, 9 NaCl, 0.9 EGTA, 9 HEPES, 14 Tris-phosphocreatine, 2 Mg-ATP, 0.3 Tris-GTP, and 2 Na2-ATP, pH adjusted to 7.3 with KOH, 290–300 mOsm. The low EGTA intracellular solution allows endogenous $Ca^{2+}$ influx to activate BK channels (*Fakler and Adelman, 2008*; *Jackson et al., 2004*; *Müller et al., 2007*; *Whitt et al., 2016*; *Whitt et al., 2018*). The slices were perfused with oxygenated ACSF at room temperature during the recordings. Granule cells in the DG were visualized with a Luca-R DL-604 EMCCD camera (Andor, Belfast, UK) under IR-DIC illumination on an FN1 upright microscope (Nikon, Melville, NY). Current- and voltage-clamp recordings were made with a Multiclamp 700B amplifier, and signals were filtered at 10 kHz and digitized at 50 kHz using Digidata1440A and pCLAMP v10 software (Molecular Devices, Sunnyvale, CA). All data were corrected for liquid junctional potential (10 mV).

In current-clamp recordings, the resting membrane potential was measured without any current input after a whole-cell mode was made. If the initial resting membrane potential was more depolarized than −70 mV or a series resistance was larger than 20 MΩ, the recording was not proceeded. The membrane potential was maintained at around −80 mV by injecting positive currents. Firing frequency was obtained from running a single protocol containing 1 s step current injections (0 to +400 pA, 20 pA increment, 10 s ISI). Bridge balance was used. The input resistance ($R_i$) was measured with a linear regression of voltage changes from 400 ms hyperpolarizing current injections (−40 to −10 pA in 10 pA increments). The membrane time constant was calculated from the averaged traces to 20 consecutive hyperpolarizing current pulses (−20 pA; 400 ms) with a single exponential function (*Lopez et al., 2012*). If the series resistance ($R_s$) or membrane capacitance ($C_m$) was changed more than 20% over the recording, the cell was not further analyzed.

In voltage-clamp recordings, $R_s$ was compensated at least 60%. BK currents were measured by subtracting currents with 10 μM paxilline from the total current in the presence of 1 μM tetrodotoxin (TTX) and 2 mM 4-aminopyridine (*Montgomery and Meredith, 2012*). Cells were held at −90 mV, 150 ms voltage step of −100 to +30 mV in 10 mV increments was applied and stepped back to −90 mV for 130 ms. Three current traces were averaged for analysis, and leak currents were subtracted using the P/4 method with a subsweep holding potential of −90 mV. Paxilline was applied to the slice using a local perfusion pencil for at least 10 min prior to the second recording. BK current levels were obtained from the peak and normalized to cell capacitance. No paxilline-sensitive current was present in *Kcnma1*$^{−/−}$ dentate granule neurons (n=2).

## Action potential waveform analysis

Action potential amplitude was defined as the difference between the peak and threshold. Half width ($t_{1/2}$) was the width of action potential at 50% of the peak amplitude from the action potential threshold. The amplitude of fAHP was defined as the voltage change from the action potential threshold to the most negative voltage (AHP anti-peak) after repolarization. The fAHP decay was measured as the depolarization rate from the AHP anti-peak over the first 3 ms ($\Delta V/3$ ms). Action potential thresholds, defined as the membrane potential where the first derivative reached 10 mV/ms, were not different between genotypes for any mouse line (data not shown).

## EEG and EMG recordings

Behavioral observations, surgeries, EEG and EMG recordings and data analyses were performed blinded to experimental condition and genotype. After daily monitoring of behavioral signs of seizures, 2- to 4-month-old mice were implanted with dural electrodes, with or without EMG lead implantation at the dorsal clavotrapezious neck muscles behind the base of the skull Pinnacle Technology 4 channel EEG system, Lawrence, KS (*Iffland et al., 2020*). Following a minimum of 72 hr of recovery period, video-EEG recordings were obtained using Pinnacle Technology Sirenia Acquisition software for 24 hr at a sampling rate of 2 kHz. Mice were visually monitored during seizures and behavioral responses were scored using modified Racine criteria: (1) raised tail and/or abnormal posturing; (2) myoclonic movement of a limb, favoring one side; (3) brief tonic-clonic convulsive episodes (approx. 1–5 s); (4) tonic-clonic seizures associated with rearing or jumping; and (5) status epilepticus (*Lüttjohann et al., 2009*; *Van Erum et al., 2019*). Continuous EEGs were manually reviewed for interictal epileptiform discharges and/or spontaneous seizures. Interictal epileptiform discharges were defined as discrete and sharply contoured discharges (e.g., spike and wave). Seizures were defined as at least 10 s of sharply contoured and rhythmic bursts of activity.

Seizure thresholds were defined in response to a single chemoconvulsant challenge in naïve animals. A baseline 15 min video-EEG recording was obtained, followed by injection of 40 mg/kg of PTZ (Sigma, Cat. #P6500, 10 mg/ml stock in sterile saline) and an additional 30 min recording after injection. Thirty minutes post-PTZ injection, the experiment was terminated, and mice were euthanized by $CO_2$ asphyxiation and thoracotomy. Video-EEG and EMG data were analyzed using Sirenia Seizure Pro (Pinnacle Technology, Lawrence, KS). Experimenters were blinded to experimental group during PTZ-induced seizure, observation, and data analysis. Seizures were defined as 10 continuous seconds of sharply contoured and rhythmic discharges with a clear onset, offset, and evolution. Seizure threshold was defined as the latency to first seizure after PTZ injection. Total EEG power was reported as the sum of all frequency bands. EEG traces were examined visually for significant artifacts, and EEGs resulting in anomalous power data were excluded from the analysis, defined as EEGs with high amplitude movement artifacts (>16,000 $\mu V^2$/Hz) or low amplitude signals (<750 $\mu V^2$/Hz). EMG data were analyzed by manual review and the longest durations of attenuated EMG activity were scored. Attenuated EMG activity was defined as at least 1 s of EMG activity that was lower in amplitude than pre-PTZ injection baseline.

## Stress-induced dyskinesia assays

Two- to 3-month-old mice were used in all assays for N999S and D434G cohorts. Two- to 8-month-old mice were used for *Kcnma1*$^{-/-}$ cohorts due to breeding difficulties (*Meredith et al., 2004*). For acute stress-induced dyskinesia evaluation, the total restraint time was 5 min. Mice were restrained for 2.5 min by hand, clasping the dorsal cervical aspect between the index finger and thumb and the tail with the pinky finger, with the mouse dorsal side flat against the palm in a vertical upright position. Afterward, the tail was released, leaving only the upper body restrained for 2.5 min. In separate experiments, dextroamphetamine (d-amp) was administered via intraperitoneal (IP) injection at 0.5 mg/kg, or an equivalent volume of saline, to a final volume of 5 ml/kg. Syringes were prepared by a second blinded experimenter. Thirty minutes after injection, mice were restrained in a plexiglass tube (MH-100, IBI Scientific, Dubuque, IA) for 5 min.

After restraint, mice were placed in a transparent 1000 ml beaker under video recording for 5 min. Behavioral parameters were modified from stereotypic behavioral scoring (*Kelley, 2001*) and prior dyskinesia mouse models (*Khan et al., 2004*; *Khan and Jinnah, 2002*; *Sebastianutto et al., 2016*; *Shirley et al., 2008*) and were manually scored from side-view videos totaled for grooming time, or the number of occurrences of rearing, sniffing, circling/hyperactivity, twisting/choreiform movement, tonic-clonic movement, flattened/dystonic posturing, tremoring, listing, and falling. Immobility/behavioral arrest was defined as lack of positional and translational movement from the body, limbs, and head, excluding involuntary movement from respiration. Episodes were cumulatively timed to obtain the total immobility in 5 min. Videos from d-amp-injected mice were additionally subjected to automated analysis (EthoVision XT 11.5, Noldus) by a third blinded experimenter. Movement tracking was calibrated from side-view video to beaker dimensions: rim = 11.110 cm, base = 10.795 cm, and height = 15.875 cm. Immobility time was obtained from the center point movement parameter with automated body detection. Mice were considered mobile with center point velocities exceeding

0.151 cm/s, and immobility time was obtained from intervals where the center point velocity was below 0.050 cm/s.

## Open field activity

Mice were acclimated in the testing room 1 hr prior to assays. Each mouse was placed in the open arena (70 × 30 × 25 cm³, *Cover et al., 2019*) for 15 min. Mouse movement and total distance was analyzed in EthoVision XT (Noldus).

## Wheel running activity

Mice were placed in housing cages with running wheels (Coulbourn Instruments) on a standard 12:12 hr light-dark cycle for 48 hr with ad libitum access to food and water. Wheel activity was measured via magnetic switches and recorded using ClockLab software (Actimetrics). Individual mouse wheel rotation counts were then quantified in 1 min bins in ClockLab software running in Matlab v6.1 (Mathworks). The following parameters were calculated for the 12 hr dark phase as average measurements: speed, maximum speed, number of activity gaps (defined as consecutive 1 min bins registering 0 rpm), activity gap duration, and maximum activity gap duration. All parameters were calculated by a custom python script (code provided as a source data file).

In separate mice, baseline wheel activity recording was collected (30 min), followed by 0.5 mg/kg d-amp or saline injections as described in the previous section. Thirty minutes after injection, each mouse's post-treatment activity count (30 min) was normalized to the baseline.

## Rotarod

Motor coordination was tested by rotarod over 7 days of trials on an accelerating rod. Mice were acclimated to the testing environment in their normal housing cages for 1 hr prior to testing. Mice were trialed three times a day for 7 consecutive days under video capture. Body weight was measured on days 1 and 7. After placement on the rotarod (IITC Life Science Inc, Rat Mouse Rotarod), mice were acclimated on the apparatus for 30 s prior to first trial. During the acclimation, mice were allowed to fall off up to two times. Rod acceleration was 4–40 rpm over 5 min for each trial. Mice were given a 2 min inter-trial interval. 'Fall' was called when the mice fell off from the rod or made a 360 degree revolution around the rod. Three trials per day were averaged for each mouse.

Hypotonia and PNKD-like paroxysms would be expected to produce extremely short latencies to fall, since a single major loss of tone would be catastrophic for coordination. Alternately, baseline coordination could be impaired in the absence of immobility. Among the individual mice, there was no explicit evidence for individual trials with extremely short fall latencies that would be consistent with the triggering of sudden hypotonic events. However, partial loss of tone might be compensated for by the high level of attention induced in this assay. Compared to the voluntary running wheel activity, successful navigation of the rod's surface and rotation requires a higher degree of attention to motor coordination than the wider home cage wheel. Thus, mice fall off the rotarod at lower speeds than the maximum speeds achieved on the voluntary running wheel.

## Hanging wire

Acute muscle strength was tested by hanging mice using their limbs. Mice were acclimated to the testing environment in their home cages for 1 hr prior, and body weights were measured prior to the start of testing. Three consecutive trials were then performed in 1 day. Mice were placed right-side-up on a standard cage lid with parallel metal bars, the lid was gently shaken three times to provoke grasping the bars, then the lid was slowly inverted to put mice in the hanging position. Trial duration maximum was 120 s. A 10 s inter-trial interval was given after falling. Mice remaining hanging up to the maximum time were inverted to right-side-up on the lid and given a 10 s interval before the next trial. Most WT mice can hang on for several minutes before losing grip and falling (*Jones and Mormède, 2000*), although each WT control cohort exhibited a wide range of latency to fall values. No sudden hypotonia was observed, and there was a spread in the range of *Kcnma1*^N999S/WT values. This observation suggests it unlikely that the hanging wire assay was a significant trigger for PNKD-like immobility in *Kcnma1*^N999S/WT mice, potentially due to the short duration compared to the full restraint stress used in *Figure 5*.

## Statistics

Sample sizes were determined based on prior datasets, except the stress-induced behavioral assay where power calculation based on pilot data indicated sample sizes of 5 animals per group (effect size d=2.29 at 0.8 power with 0.05 α; G*power 3.1). Electrophysiology and behavioral data were tested for normality with Shapiro-Wilk normality test and either parametric or non-parametric statistics were analyzed in GraphPad Prism v9.02 (San Diego, CA). Outliers were determined by the ROUT method and were included in all datasets. Data are plotted as either mean ± SEM, or individual data points with median and inter-quartile range, as indicated in figure legends. The statistical test used for each dataset is indicated in the figure legend, and $p<0.05$ was considered significant. p-Values in figure legends are reported for post hoc tests when the main effect was $p<0.05$, or reported for the main effect, if $p>0.05$. Groups with three or fewer data points were not included in statistical analysis, as noted in legends. For parametric data, two-tailed, unpaired t-tests were performed with Welch's correction for unequal variance. For multiple comparisons, one-way ANOVAs were performed with Welch's correction followed by Dunnett's T3 post hoc test. Two-way repeated measures ANOVAs were performed with Geisser-Greenhouse correction followed by Bonferroni post hoc test for multiple comparisons (comparisons between genotypes across voltages). Mann-Whitney or Kruskal-Wallis followed by Dunn's multiple comparisons were used for non-parametric data.

In gene microarray studies, differential mRNA transcript expression was determined at a 2-fold change cutoff, with $p>0.05$ and false discovery rate, FDR = 0.99 using an ANOVA with an eBayes test was used (*Ritchie et al., 2015*) (Applied Biosystems Transcriptome Analysis Console (TAC) Software v4.0.1).

## Acknowledgements

This work was supported by grants from NHLBI HL102758 (ALM), The Training Program in Integrative Membrane Biology NIGMS T32-GM008181 (ALM and KKM), The American Physiological Society's Ryuji Ueno award sponsored by the S&R Foundation (ALM), The Interdisciplinary Training Program in Muscle Biology NIAMS T32-AR007592 (SP), and NINDS NS114122 (PBC). For gene expression studies, we acknowledge the support of the University of Maryland, Baltimore, MD, Institute for Clinical & Translational Research (ICTR voucher #376) and the National Center for Advancing Translational Sciences (NCATS) Clinical Translational Science Award (CTSA) grant number 1UL1TR003098, Nick Ambulos and Jing Yin for performing microarray experiments, and Yuji Zhang for biostatistical analysis. We thank Sotirios Keros for helpful discussions and Huanghe Yang for discussions involving unpublished data. We thank Todd Gould and Brian N Mathur for providing use of equipment for mouse motor assays and helpful discussions and Ria Dinsdale for assistance with blinded dextroamphetamine experiments.

## Additional information

### Funding

| Funder | Grant reference number | Author |
| --- | --- | --- |
| National Heart, Lung, and Blood Institute | NHLBI HL102758 | Andrea Meredith |
| National Institute of General Medical Sciences | NIGMS T32-GM008181 | Katia K Matychak<br>Andrea Meredith |
| The American Physiological Society | Ryuji Ueno Award | Andrea Meredith |
| National Institute of Allergy and Infectious Diseases | NIAMS T32-AR007592 | Su Mi Park |
| National Institute of Neurological Disorders and Stroke | NINDS NS114122 | Peter B Crino<br>Philip H Iffland II |

| Funder | Grant reference number | Author |
| --- | --- | --- |
| National Center for Advancing Translational Sciences | 1UL1TR003098 | Andrea Meredith |

The funders had no role in study design, data collection and interpretation, or the decision to submit the work for publication.

## Author contributions

Su Mi Park, Conceptualization, Data curation, Formal analysis, Funding acquisition, Investigation, Methodology, Project administration, Resources, established the transgenic lines, performed HEK cell recordings, and neuronal recordings., Supervision, Validation, Visualization, Writing – original draft, Writing – review and editing; Cooper E Roache, performed locomotor assays, data analysis, and animal husbandry., Conceptualization, Data curation, Formal analysis, Investigation, Methodology, Project administration, Software, Validation, Visualization, Writing – original draft; Philip H Iffland, Conceptualization, Data curation, Formal analysis, Investigation, Methodology, performed EEG recordings., Project administration, Resources, Software, Supervision, Validation, Visualization, Writing – original draft, Writing – review and editing; Hans J Moldenhauer, Conceptualization, Data curation, Formal analysis, performed HEK cell recordings with d-amp and lis, animal husbandry, and provided electrophysiology analytical methodology., Investigation, Methodology, Project administration, Resources, Supervision, Validation, Visualization, Writing – review and editing; Katia K Matychak, Conceptualization, Data curation, Formal analysis, Investigation, assisted with EEG recordings and animal husbandry, Validation, Visualization, Writing – review and editing; Amber E Plante, established the transgenic lines and performed animal husbandry, Investigation, Methodology, Project administration, Validation, Writing – review and editing; Abby G Lieberman, performed open field assays and Ethovision analysis., Investigation, Methodology, Project administration, Validation, Writing – review and editing; Peter B Crino, Funding acquisition, Investigation, provided EEG recording set ups., Project administration, Resources, Supervision, Writing – review and editing; Andrea Meredith, provided KCNMA1 sequences, patient data, designed the transgenic models, and provide analytical methods in electrophysiology and motor function., Conceptualization, Data curation, Formal analysis, Funding acquisition, Investigation, Methodology, Project administration, Resources, Supervision, Validation, Visualization, Writing – original draft, Writing – review and editing

## Author ORCIDs

Katia K Matychak http://orcid.org/0000-0001-5094-5449
Andrea Meredith http://orcid.org/0000-0003-1061-2302

## Ethics

All experiments were conducted in accordance with the University of Maryland School of Medicine Animal Care and Use Guidelines and approved by the Institutional Animal Care and Use Committee (Protocol #1120011).

## Decision letter and Author response

Decision letter https://doi.org/10.7554/eLife.77953.sa1
Author response https://doi.org/10.7554/eLife.77953.sa2

## Additional files

### Supplementary files

• Transparent reporting form

• Source code 1. Locomotor wheel activity script. Python code to calculate wheel speed and the number and duration of activity gaps.

### Data availability

All data generated and analyzed during this study are included in the manuscript, or provided as source data files. Python code is provided as Source Code File 1.

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
