## [Editor Report]

This study is of broad interest to neuroscientists interested in membrane excitability and translational biologists and physicians eager for robust animal models for disorders involving mutations in the KCNMA gene, such as paroxysmal nonkinesigenic dyskinesia PNKD3. Here, phenotypes of mouse models of three of the more common patient disease-related mutations in KCNMA are evaluated for similarities to patient phenotypes. This work establishes that BK channel mutations linked to human neurological disease can, on their own, cause similar pathology in mice, and it also begins to provide neurological bases for the associated behavioral deficits. Importantly, one of the mutant alleles expressed in mice most closely phenocopies the patient phenotype, which is rescued with a new treatment for PNKD3 in KCNMA1-N999S patients, further validating it as an important animal model for studies seeking therapeutic treatments for the resulting debilitating disease moving forward.

---

## [Decision Letter]

**Decision letter after peer review:**

Thank you for submitting your article "BK channel properties correlate with neurobehavioral severity in three *KCNMA1*-linked channelopathy mouse models" for consideration by *eLife*. Your article has been reviewed by 3 peer reviewers, including Teresa Giraldez as Reviewing Editor and Reviewer #1, and the evaluation has been overseen by Richard Aldrich as the Senior Editor.

Essential revisions:

All reviewers agreed that these results are interesting and compelling. However, reviewers also raised several concerns about the interpretation of the data that should be addressed by the authors. Essential revisions should include:

1) The gene dosage effect is not fully supported or discussed. Related to this, the assumption that in transgenic neurons all channels are heteromers, or to what extent this is the case, is not sufficiently discussed.

2) Nature of the native BK currents in transgenic neurons: what is the influence of different splicing variants, or regulatory subunits? Related to this question, two reviewers point out the apparent lack of effect of the mutations on Vhalf values in transgenic neurons, which should be also addressed.

3) Further discussion is needed about how increased BK current would enhance neuronal excitability and how this might it lead to the PKND phenotype.

4) Further discussion is needed to explain why the H444Q transgenic neurons produce WT-like currents and no alterations in excitability. As mentioned by one reviewer, could the LOF allele be associated with an upregulation of the WT allele in neurons, resulting in the normal phenotype?

5) Although the study seems rigorous, in some cases the sample sizes were rather low (e.g., Figure 4A-B, data H444Q/H444Q). In such cases, the number of experiments should be increased, or alternatively not shown.

6) Text should be revised according to all comments raised by the reviewers and listed in the individual reviews below.

Because the individual reviews include several important points they are included here for your reference.

*Reviewer #1 (Recommendations for the authors):*

The manuscript is well written and structured. Data are solid and well presented. Some comments are detailed below.

P8, line 165: "…gene expression was analysed from hippocampus and cerebellum of Kcnma1-N999S/WT and WT littermates. No significant differences were found in the levels…" Does this refer to expression of Kcnma1 WT, mutant, or both? What is the conclusion of this experiment?

What are the expression levels of WT vs mutants? Do the proportions change when mutants and WT are co-expressed in HEK cells?

In experiments from Figure 2 and in general, in all data from heterozygous animal models, it appears that the authors are assuming that the currents correspond to mutant/WT heteromers. But this is not necessarily the case. This aspect should be better discussed. This is especially relevant in the case of the H444Q mutant (see below).

Figure 2C: H444Q/WT neurons produce similar currents than WT. Have the authors tested if H444Q mutants are being expressed in these neurons? Could these currents be produced just by WT? (non-significant differences are found in all other studied functional features, in Figure 3 and 4) Have the authors tested co-expression of H444Q/WT combinations in HEK cells?

What are the values for the BK G-V shifts observed in neurons from N999S/WT, D434G/WT and D434G/D434G mice?

Figure 4: The low number of experiments with homozygous D434G or H444Q is so low that as the authors remark it is impossible to obtain any conclusion. Is there any reason for this low number of experiments? If no more data are available, what is the purpose of showing inconclusive results here?

Figure 5A show data corresponding to control vs stress conditions for WT vs N999S mice. Panels C-E show data corresponding to stress conditions for the other mutants. How is the comparison of D434G and H444Q mice with and without stress conditions? In other words, do these mutants show any significant difference in mobility in basal conditions when compared to the WT?

*Reviewer #2 (Recommendations for the authors):*

1) lines 184-195, Figure 2: The GOF mutants show an increase in the size of neuronal BK currents, but there is no apparent shift in the voltage-activation range for the mutants compared to WT, as one might expect based on the HEK recordings. So one wonders if the neuronal BK currents have been well-isolated, or if the presence of the BK-beta4 subunit in DG neurons decreases the shift that one might expect for the mutant currents. In either case, this should at least be addressed with some discussion.

2) lines 257-260 "Analysis of waveforms from the 200 pA step corroborate BK channels regulate multiple phases of the action potential, but suggested the basis for increased firing in Kcnma1N999S/WT and Kcnma1D434G/D434G neurons was a faster AHP decay rate (Figure 3‒figure supplement 2), which would facilitate more rapid initiation of the next action potential."

As an explanation for the relation between the effects of these mutations on BK currents and their consequences on AP firing, this seems a bit counterintuitive. Figure 1 (panels C and G) shows that in HEK cells, N999S and D434G deactivate more slowly than WT. So wouldn't one expect these effects to lead to a slower AHP decay rate at negative voltage compared to WT? Also, is it known whether these mutations lead to different effects on BK gating kinetics when co-expressed with the beta4 subunit, or whether they may have different effects on action potential shape or AHP in other neurons? ? This may be worth a little more discussion.

*Reviewer #3 (Recommendations for the authors):*

A few other questions arose that could be interesting if addressed in the Discussion.

1. For example, what is the role of mutant muscle BK currents in disease phenotype?

2. Is there a possibility that the differences among GOF mutant phenotypes could be attributable to the splice variant context in different brain regions? Is this worth discussing?

3. Future studies might test whether the LOF allele is associated with an upregulation of the WT allele in neurons, resulting in the normal phenotype and possibly pointing to mechanisms accessible to rescue.

---

## [Author Response]

Essential revisions:All reviewers agreed that these results are interesting and compelling. However, reviewers also raised several concerns about the interpretation of the data that should be addressed by the authors. Essential revisions should include:1) The gene dosage effect is not fully supported or discussed. Related to this, the assumption that in transgenic neurons all channels are heteromers, or to what extent this is the case, is not sufficiently discussed.

The central theme of the manuscript concerns comparisons between N999S, D434G, and H444Q. The gene dosage effect was an argument only for the D434G variant, comparing *Kcnma1^D434G/WT^* and *Kcnma1^D434G/D434G^*. While the data were stated as not conclusive in the original submission, the section and figures were revised. The *Kcnma1^D434G/D434G^* data in Figure 4Aii/Bii was removed, and the statement that data ‘suggested’ a gene dosage effect was changed to the data ‘*raise the possibility of a gene dosage effect’* with D434G (lines 388; see also the response to reviewer #3). We agree that the idea that the *Kcnma1^D434G/D434G^* seizure phenotype is more severe would be better supported with additional experiments in the future.

On the other hand, the increased severity of *Kcnma1^D434G/D434G^* PNKD-immobility is fully supported by the data with sufficient statistical power (Figure 5D).

In the second part of this concern, to address the overall issue about assumptions of heterotetramers, two additional paragraphs were added to the Results and Discussion sections. Details are found in the responses to individual reviewers that follow below.

2) Nature of the native BK currents in transgenic neurons: what is the influence of different splicing variants, or regulatory subunits? Related to this question, two reviewers point out the apparent lack of effect of the mutations on Vhalf values in transgenic neurons, which should be also addressed.

None of the variants are located in regions expected to alter expression, splicing, or β subunit association. Our characterization was limited to assessing BK current levels, due to the whole-cell slice recording configuration. The data provide guidance for interpretation on the questions of how splicing or regulatory subunits may affect current properties, but cannot inform a definitive mechanism. In tandem with the heterotetramer formation in the previous question above, whether splicing and β subunit regulation contribute to lack of change in V_1/2_s, is addressed in the additional paragraphs added to the Results and Discussion sections, with specific details within the responses to individual reviewers that follow.

3) Further discussion is needed about how increased BK current would enhance neuronal excitability and how this might it lead to the PKND phenotype.

In general, this is not well understood for any type of episodic dyskinesia at present. Our study represents a first step toward this goal for *KCNMA1*-associated PNKD3. In this first step, we focus on cross-comparison between several PNKD-associated variants, rather than a deeper mechanistic dive into its basis. To our knowledge, the *Kcnma1^N999S/WT^* mice are the only model that shows episodic, triggered hypotonic dyskinesia (PNKD3), among the very few PNKD models that are currently characterized. New data showing that dextroamphetamine can rescue the PNKDimmobility in *Kcnma1^N999S/WT^* mice (new data in Figure 5C) provides the speculation for a potential CNS basis. Two paragraphs have been added to address the respective mechanistic questions posed here (added at the end of the action potential section, lines 294-304, and in the Discussion, lines 675-700), and detailed responses are provided within the individual reviewer comments.

4) Further discussion is needed to explain why the H444Q transgenic neurons produce WT-like currents and no alterations in excitability. As mentioned by one reviewer, could the LOF allele be associated with an upregulation of the WT allele in neurons, resulting in the normal phenotype?

In heterologous cells, BK^H444Q^ produced the mildest effects on BK channel properties as mutant homotetramers. We added additional paragraphs to the Results section discussing BK current levels (lines 210-231) and Discussion (655-674) outlining the factors that could influence the phenotypes (or lack thereof) observed in *Kcnma1^H444Q/ WT^* and other heterozygous transgenic conditions. More detailed responses, including preliminary data from BK^H444Q/WT^ recordings are provided in responses to Reviewer #1.

5) Although the study seems rigorous, in some cases the sample sizes were rather low (e.g., Figure 4A-B, data H444Q/H444Q). In such cases, the number of experiments should be increased, or alternatively not shown.

*Kcnma1^H444Q/ H444Q^* data were inadvertently included in Figure 4Aiii and Biii. There was no conclusion in the text from this data, and it was not included in statistical analysis. The figure has been corrected.

6) Text should be revised according to all comments raised by the reviewers and listed in the individual reviews below.

See responses that follow to specific comments raised by all reviewers. In addition, to more clearly define the scope of the study, a reference was added after the central hypothesis, citing MacArthur et al., 2014 (‘Guidelines for investigating causality of sequence variants in human disease;’ lines 64 and 574).

Because the individual reviews include several important points they are included here for your reference.Reviewer #1 (Recommendations for the authors):The manuscript is well written and structured. Data are solid and well presented. Some comments are detailed below.P8, line 165: "…gene expression was analysed from hippocampus and cerebellum of Kcnma1-N999S/WT and WT littermates. No significant differences were found in the levels…" Does this refer to expression of Kcnma1 WT, mutant, or both? What is the conclusion of this experiment?

We added the following text to the section to clarify:

“The results show that no gross up- or down-regulation of BK channel components occurs in either the hippocampus or cerebellum of Kcnma1^N999S/WT^ neurons. However, since the microarray-based probe set does not distinguish between the WT and N999S Kcnma1 transcripts, no further conclusion can be made regarding the specific expression ratio of each allele.”(lines 169-173).

What are the expression levels of WT vs mutants? Do the proportions change when mutants and WT are co-expressed in HEK cells?

See the previous response clarifying that WT versus N999S expression levels cannot be obtained from the commercially-available microarray probe set we used. The levels were not assessed for the other mouse lines.

If changes in transcript levels did occur, we think they would be more likely to occur with the strongest mutation affecting channel properties (N999S). For N999S, the lack of change in total *Kcnma1* levels in two mouse brain regions does not provide an immediate reason to invoke changes in the ratio. We also don’t have reason to think that the CRISPR-introduced single base changes alter allelic expression ratios, now added as an additional comment on whether expression might affect the BK current levels observed in the study (lines 213-216).

We have limited data from co-expressed WT/N999S and WT/H444Q channels in HEK cells (see answers to the related question below also). The results are what would be expected: coexpressed channels have intermediate phenotypes for V_1/2_. However, looking more closely at individual values reveals bimodal distributions in both cases. Additional work would be required to determine whether this distribution is produced from WT, WT:mut, and Mut-only channels.

However, HEK cells are not an adequately quantitative system to address the issues of the ratio of WT and mutant allele expression. If the WT and mutant plasmids were transfected in a 1:1 ratio, mimicking a de novo heterozygous allele configuration, the plasmid promotor regions still lack genomic regulatory control sequences. Moreover, in a recent study addressing a similar issue, the effect of co-expressing WT BK channels and G375R mutant channels was assessed on stoichiometry (Geng et al., BioRxiv 2021), added to lines 223-225. In that study, 85% of the channels had at least 1 mutant subunit, with 12% containing all mutant or 3% all WT subunits. Yet knowing this from heterologous cells still leaves completely open how different neuronal types would manifest the same experiment, given the elaboration of ion channel trafficking and subcellular localization mechanism.

In experiments from Figure 2 and in general, in all data from heterozygous animal models, it appears that the authors are assuming that the currents correspond to mutant/WT heteromers. But this is not necessarily the case. This aspect should be better discussed. This is especially relevant in the case of the H444Q mutant (see below).

New paragraphs addressing this concern have been added to the end of the Results section containing the data on BK currents in dentate granule neurons for each transgenic line (lines 210-231). The paragraph also incorporates additional aspects of the functional effects on BK currents raised in other questions below and from the other reviewers. Another new paragraph in the Discussion places the lack of understanding in this area in the context of how to think about phenotypic severity (lines 658-674).

Figure 2C: H444Q/WT neurons produce similar currents than WT. Have the authors tested if H444Q mutants are being expressed in these neurons? Could these currents be produced just by WT? (non-significant differences are found in all other studied functional features, in Figure 3 and 4) Have the authors tested co-expression of H444Q/WT combinations in HEK cells?

Whether both alleles are expressed was not directly tested in Kcnma1^H444Q/WT^, but we added some interpretation into the Discussion that indirectly suggests the variant-containing alleles are expressed (lines 646-649):

“It is likely that all 3 variant-containing alleles are expressed because the homozygous phenotypes are not similar to Kcnma1^—/—^: Kcnma1^N999S/N999S^ and Kcnma1^D434G/D434G^ have unique phenotypes not found in Kcnma1^—/—^ (lethality and stress-induced immobility, respectively), and Kcnma1^H444Q/H444Q^ has fewer and less severe phenotypes than Kcnma1^—/—^ (e.g. rotarod).”

Though this doesn’t preclude changes in expression levels, as discussed in the prior response, or directly speak to the heterozygous cases, it is at least an indirect indicator that lack of variant allele expression is not likely to be a primary driver of the phenotypes observed in the study.

We have limited data from co-expressed WT/H444Q channels in HEK cells (see answers to the related question previously). Co-expressed WT/H444Q channels have intermediate phenotypes for V_1/2_. We also have a limited number of recordings ± β4. This data suggests that BK^WT+H444Q^, either with or without β4, is not distinguishable from BK^WT^+β4 in HEK cell recordings. New additions to the manuscript provide more discussion on the potential for β4 to be a key player in mitigating differences in Kcnma1^H444Q/WT^ BK current properties, given its high levels in dentate granule neurons (see also lines 229-231). Future studies dissecting the expression, subunits, and localization for variant-containing channels could reveal a more detailed level of mechanism.

What are the values for the BK G-V shifts observed in neurons from N999S/WT, D434G/WT and D434G/D434G mice?

The apparent voltage-dependence calculated from fits of the average I/I_max_ were not different for any of the transgenic conditions tested. They were:

Kcnma1^WT/WT^ (4.3±1.2 mV) and Kcnma1^N999S/WT^ (6.5±1.8 mV)

Kcnma1^WT/WT^ (4.7±1.9 mV) and Kcnma1^D434G/WT^ (5.2±3.5 mV), Kcnma1^D434G/D434G^ (6.0±2.5 mV) Kcnma1^WT/WT^ (4.4±0.5 mV) and Kcnma1^H444Q/WT^ (5.6±2.0 mV)

The factors that may influence the lack of observed changes are discussed in more detail in two new paragraphs (mentioned in the prior response). Added to the Results (lines 210-231), specifically:

“Several factors that could mitigate differences in V_1/2_ are undefined, including the α

(WT:mutant) and β subunit stoichiometry, splice variant background, and intracellular ca^2+^. Limited data is available addressing some of these. Co-expression of WT and mutant (GOF) BK channel cDNAs supports the assumption that heterotetramers are the predominant channel type produced by 1:1 transcript ratios in *Xenopus oocytes* (Geng et al., 2021). A few studies have shown that N999S and D434G confer similar ∆V_1/2_ onto different splice variants (Figure 1C-G; Li et al., 2018; Moldenhauer et al., 2020; Wang et al., 2009) and maintain left-shifted V_1/2_ values compared to WT in the presence of the β4 subunit (Berkefeld and Fakler, 2013; Li et al., 2018; Wang et al., 2009). Yet with D434G, less of a difference ± β4 is found above 10 µM Ca^2+^ (Wang et al., 2009), which could be significant in granule neurons given the widespread abundance of β4.”

Added to the Discussion (lines 658-674):

“At the channel level, unresolved questions concerning functional mechanisms and subunit composition illustrate the influence of additional factors besides variant genotype on the phenotypic severity. For example, BK current levels were relatively similar between Kcnma1^N999S/WT^ and Kcnma1^D434G/WT^ in one type of neuron (dentate granule), yet these two heterozygous genotypes exhibited differences in excitability. Instead, homozygous Kcnma1^D434G/D434G^ firing curves were more comparable to Kcnma1^N999S/WT^. Underlying this, the ratio of expression from WT and mutant alleles, stoichiometry and properties of heterotetrameric channels, alternative splice variation, and the composition of α:β:CaV macrocomplexes in the loci responsible for neuropathology in heterozygous transgenic mice all still remain to be resolved. Additional experiments to probe the voltage and Ca^2+^ dependent bases for N999S and D434G gating defects (Diez-Sampedro et al., 2006; Du et al., 2005; Li et al., 2018; Moldenhauer et al., 2020; Yang, 2010), and the manifestation in the presence of the highly-expressed β4 subunit (Wang et al., 2009; Berkefeld and Fakler, 2013; Wang et al., 2016), may further explain these relative differences in BK currents and pathogenicity. Interestingly, no KCNMB4 variants associated with seizure have been described yet, even as the number of seizure-associated BKα variants has increased (Bailey et al., 2019; Miller et al., 2021). This could suggest that loss of β4 regulation could have more severe consequences.”

Figure 4: The low number of experiments with homozygous D434G or H444Q is so low that as the authors remark it is impossible to obtain any conclusion. Is there any reason for this low number of experiments? If no more data are available, what is the purpose of showing inconclusive results here?

*Kcnma1^H444Q/H444Q^* data were inadvertently included in Figure 4Aiii and Biii. There was no conclusion in the text from this data, and it was not included in statistical analysis. The figure and figure legend have been corrected.

The *Kcnma1^D434G/D434G^* data have also been removed from the figure itself, addressing a concern of reviewer #3 as well. The individual measurements themselves have been transferred to the Results text (lines 385-387). We think the *Kcnma1^D434G/D434G^* seizure threshold and power values merit inclusion as individual measurements in order to make direct comparisons with other independent D434G studies. Because the three measurements in our dataset were consistent, a trend toward the shortest seizure latencies is suggested. This provides a basis for comparison to an independently generated D434G mouse model (Dong et al., 2022). We hope removal from the figure still allows for cross-comparability with other datasets, without overemphasizing a conclusion based on a dataset comprised of 3 measurements.

Figure 5A show data corresponding to control vs stress conditions for WT vs N999S mice. Panels C-E show data corresponding to stress conditions for the other mutants. How is the comparison of D434G and H444Q mice with and without stress conditions? In other words, do these mutants show any significant difference in mobility in basal conditions when compared to the WT?

Because the experiments without restraint stress and with restraint stress are conducted on separate cohorts, we did not have enough *Kcnma1^D434G/D434G^* and *Kcnma1^H444Q/ H444Q^* animals to perform this control for every mouse line. However, the construct validity for the PNKD assay is further supported by new data in panel 5C. The new data show that restraint stress conducted with a modification to the original methodology (using a plexiglass restraint, after injecting the mice, and analyzed by two independent experimenters using separate manual and automated analysis methods) still cannot induce significant immobility in any WT animals.

To answer the second question, Figure 6 supplements 1 and 2 contain experiments addressing baseline mobility differences for D434G and H444Q mice. These data show some aspects of reduced motor coordination (6-1Ci and 6-2Bi *Kcnma1^D434G/D434G^* and 6-2Aii *Kcnma1^H444Q/ H444Q^*). However, these baseline defects are not likely to affect the immobility observed in the stress-induced dyskinesia assay, which is conducted in a beaker (10 cm diameter). Details for the baseline motor phenotypes were added to lines 553-554 (*Kcnma1^D434G/D434G^*) and to lines 561-566 (*Kcnma1^H444Q/ H444Q^* and *Kcnma1^—/—^*). We have also performed gait analysis under basal conditions using CatWalk (data not shown). Based all these assays, we do not think there are any confounding ataxic or hypotonic phenotypes in any of the variant-containing transgenic lines that would caveat the interpretations for the stress-induced dyskinesia assay in the beaker.

Reviewer #2 (Recommendations for the authors):1) lines 184-195, Figure 2: The GOF mutants show an increase in the size of neuronal BK currents, but there is no apparent shift in the voltage-activation range for the mutants compared to WT, as one might expect based on the HEK recordings. So one wonders if the neuronal BK currents have been well-isolated, or if the presence of the BK-beta4 subunit in DG neurons decreases the shift that one might expect for the mutant currents. In either case, this should at least be addressed with some discussion.

In the whole-cell slice recording context, our conclusions are limited to comparisons between BK current levels. The biophysical properties and their underlying mechanisms are more advanced topic that requires a different study design. It is correct that the apparent voltage dependence derived from fits of the average I/I_max_ are not different for any of the transgenic conditions tested (see response to Reviewer #1). To address this, a new paragraph identifying the potential influence of α (WT:mutant) stoichiometry, β subunits, splice variant background, and intracellular ca^2+^ has been added at the end of the section reporting the neuronal BK current levels (lines 210-231). There is not enough cross-comparable data in the literature (collected under conditions relevant to those employed in this study) to speculate which mechanism may contribute the most to the lack of difference observed in V_1/2_ values, but we agree that the possibility that the highly expressed β4 subunit would mitigate some of these differences should be added into the discussion. These ideas are also brought back into the summaries on channel and neuronal factors that contribute to phenotypic variability in the re-organized Discussion (lines 658-674).

2) lines 257-260 "Analysis of waveforms from the 200 pA step corroborate BK channels regulate multiple phases of the action potential, but suggested the basis for increased firing in Kcnma1N999S/WT and Kcnma1D434G/D434G neurons was a faster AHP decay rate (Figure 3‒figure supplement 2), which would facilitate more rapid initiation of the next action potential."As an explanation for the relation between the effects of these mutations on BK currents and their consequences on AP firing, this seems a bit counterintuitive. Figure 1 (panels C and G) shows that in HEK cells, N999S and D434G deactivate more slowly than WT. So wouldn't one expect these effects to lead to a slower AHP decay rate at negative voltage compared to WT? Also, is it known whether these mutations lead to different effects on BK gating kinetics when co-expressed with the beta4 subunit, or whether they may have different effects on action potential shape or AHP in other neurons? ? This may be worth a little more discussion.

Without knowledge of the additional factors discussed in prior questions (heterotetramer formation, β subunits, etc), the data from this study are insufficient to make straight-forward correlations concerning BK channel and action potential properties. Our study design was limited to the characterizing the action potential waveforms during increased firing. To address the concern and perceived relationship between the properties linked to homotetrameric mutant BK channels and the transgenic neuronal action potentials, we added a new paragraph at the end of the Results section reporting action potential effects (lines 294-304). The paragraph more clearly states that mechanism by which GOF BK channels facilitate AHP decay is not revealed in this study. Though the N999S and D434G variants both slow deactivation in heterologous cells, the deactivation rate remains to be defined in *Kcnma1*^N999S/WT^ and *Kcnma1*^D434G/D434G^ neurons under repetitive firing conditions. Additional recordings under clamped Ca^2+^ directly measuring deactivation in *Kcnma1^D434G/WT^*, *Kcnma1^D434G/D434G^*, *Kcnma1^N999S/WT^*, and the respective control dentate granule neurons would be necessary. However, the slowed deactivation of mutant channels versus the accelerated AHP decay is an interesting paradox that can be informed by some prior studies with deletion of the β4 subunit. Loss of β4 produces increased BK channel open probability, fast gating, and is associated with similar changes in the action potential. These details are elaborated in the new paragraph.

Reviewer #3 (Recommendations for the authors):A few other questions arose that could be interesting if addressed in the Discussion.1. For example, what is the role of mutant muscle BK currents in disease phenotype?

This is an interesting and salient question, and one where we don’t have definitive data from patients that inform our thinking. One patient harboring the N999S variant exhibited hypotonia at birth, but underwent EMG and NCV testing as a child, which were both normal. So far, I am aware of only this single patient (who also harbors additional genetic findings) who has undergone this testing. New genotype-phenotype observations from the growing patient population will hopefully shed further light on this issue.

At the level of the BK channel, strong GOF mutations such as N999S show significant subthreshold current using a CNS action potential stimulus in 10 µM intracellular ca^2+^ (and to a much lesser extent with D434G; Moldenhauer et al., 2019). This raises the possibility that GOF mutations could hyperpolarize skeletal muscle to decrease contraction, contributing to hypotonia. We previously opened the possibility that hypotonia could manifest in several of the motor assays we conducted, such as open field (which is normal, lines 518; Figure 6B), running wheel activity (which shows increased gaps, but no change in peak speed, lines 520-521; Figure 6Ai-Aiv), rotarod and hanging wire (which were both abnormal; line 540-541; Figure 6C-D). Thus hypotonia cannot be definitively ruled out.

Underscoring the complexity of this question, both these GOF lines and Kcnma1^—/—^ (LOF) show behavioral indicators of muscle weakness (Figure 6C, Figure 6—figure supplement 2A), which in the case of Kcnma1^—/—^ is associated with decreased NMJ transmission (Wang et al., 2020). Whether this also occurs in Kcnma1^N999S/WT^ and Kcnma1^D434G/D434G^ is best answered by recording NMJ transmission and muscle V_m_/contraction directly in a future study.

D-amp rescue of the stress-induced dyskinesia also indirectly suggests hypotonia is not a major contributor to the immobility, as it would not be d-amp responsive at the low dose used in the study. This is summarized as a future direction in the expanded neurobehavioral Discussion paragraph:

“Clinical observations and d-amp responsiveness in patients predict a central neurological dysfunction in producing the debilitating hypokinetic state (Keros et al., 2022; Zhang et al., 2020), but whether other factors such altered neuromuscular transmission or episodic muscle hypotonia make some contribution during attacks remains to be systematically tested.” (Lines 696-697).

2. Is there a possibility that the differences among GOF mutant phenotypes could be attributable to the splice variant context in different brain regions? Is this worth discussing?

Splice variation in the brain has the potential to alter neuronal BK current properties from those predicted in heterologous cells, lines 221-222. In addition, more details of mechanisms that would contribute to differences in phenotypic severity were added to the Discussion. Already mentioned in this section was the differing Ca^2+^ and voltage dependent mechanisms between N999S and D434G, but splice variation is now added in this paragraph as well (line 665).

3. Future studies might test whether the LOF allele is associated with an upregulation of the WT allele in neurons, resulting in the normal phenotype and possibly pointing to mechanisms accessible to rescue.

We think the lack of phenotype in H444Q mice is due to the smaller effect on channel properties, as well as the possibility of contribution from other mutations that explain the severity of the patient phenotype. Although we don’t think the single nucleotide modifications for any of the variants are likely to change WT allele expression (line 215-216), it remains a formal possibility, given the lack of biophysical data from neuronal recordings to rule it in or out through electrophysiology. There are a few examples of heterozygous phenotypes in the *Kcnma1^—/+^* (aka Slo het) mice, so this suggests there are at least some cases where loss of the 2^nd^
*Kcnma1* allele is compensated (presumably that would occur from the WT allele), but it hasn’t been studied directly. As suggested in the question, such a compensatory mechanism could be important for rescuing LOF-related pathophysiology. Delivery of WT BK channel cDNAs has already been tested in human patients with urinary incontinence and erectile dysfunction (Melman et al., 2005 and 2006), suggesting the potential success for such a treatment in other paradigms such as congenital mutations.